# Disentangling Shared and Private Neural Dynamics with SPIRE: A Latent Modeling Framework for Deep Brain Stimulation

## Abstract

Disentangling shared network-level dynamics from region-specific activity is a central challenge in modeling multi-region neural data. We introduce **SPIRE** (Shared–Private Inter-Regional Encoder), a deep multi-encoder autoencoder that factorizes recordings into shared and private latent subspaces with novel alignment and disentanglement losses. Trained solely on baseline data, SPIRE robustly recovers cross-regional structure and reveals how external perturbations reorganize it. On synthetic benchmarks with ground-truth latents, SPIRE outperforms classical probabilistic models under nonlinear distortions and temporal misalignments. Applied to intracranial deep brain stimulation (DBS) recordings, SPIRE shows that shared latents reliably encode stimulation-specific signatures that generalize across sites and frequencies. These results establish SPIRE as a practical, reproducible tool for analyzing multi-region neural dynamics under stimulation.

## 1 Introduction

Understanding how distributed brain regions coordinate—and how this coordination is reorganized by interventions such as deep brain stimulation (DBS)—remains a major challenge. Disorders like dystonia and Parkinson's involve dysfunction in basal ganglia–thalamo–cortical circuits (Galvan et al., 2015; Jinnah & Hess, 2006; Obeso et al., 2008; Zhuang et al., 2004), and while DBS of targets such as globus pallidus internus (GPi) and subthalamic nucleus (STN) is clinically effective (Benabid, 2003; Lozano et al., 2019; Larsh et al., 2021) its network-level mechanisms remain poorly understood.

Most DBS analyses focus on local features (e.g., spectral power, evoked potentials) (Dale et al., 2022; Milosevic et al., 2018; Anonymous, 2025; 2024b), yet growing evidence shows that DBS reshapes cross-regional dynamics (McIntyre & Hahn, 2010; Horn & Fox, 2020; Yang et al., 2021; Schmidt et al., 2020). Latent variable models can capture such effects by reducing neural activity to low-dimensional subspaces, but existing methods have key limitations. Classical models such as Gaussian Process Factor Analysis (GPFA) (Yu et al., 2008) and Canonical Correlation Analysis (CCA) (Bach & Jordan, 2005) assume linearity. DLAG (Delayed Latents Across Groups) (Gokcen et al., 2022) disentangles shared vs. private dynamics but is restricted to linear–Gaussian structure and spiking data. Multimodal models (SharedAE (Yi et al.), MMVAE (Shi et al., 2019)) align shared spaces but are not designed for intracranial recordings under stimulation.

Critically, none of these frameworks provide a nonlinear, disentangling model that can separate shared versus private dynamics in human local field potential (LFP) data under external perturbation. Addressing this gap is essential: understanding how stimulation reorganizes intrinsic cross-regional coordination could reveal circuit-level mechanisms of DBS that remain invisible to local analyses.

We introduce **SPIRE (Shared–Private Inter-Regional Encoder)**, a deep multi-encoder autoencoder designed to model cross-regional dynamics under DBS. By training only on off-stimulation data, SPIRE establishes a baseline model of intrinsic coordination, which can then be probed under DBS to reveal stimulation-induced reorganization. SPIRE combines nonlinear sequence modeling with novel alignment and disentanglement losses, enabling robust recovery of cross-regional structure even under temporal misalignments and nonlinear distortions.

Applied to synthetic benchmarks with ground-truth latents, SPIRE outperforms classical probabilistic models in recovering shared and private processes under realistic conditions. Applied to pediatric intracranial DBS recordings, SPIRE shows that shared latents reliably encode stimulation-specific signatures that generalize across sites and frequencies. These findings provide both methodological and neuroscientific contributions: a reproducible framework for analyzing multi-region neural data, and, to our knowledge, the first demonstration that disentangled latent modeling reveals frequency-dependent reorganization of basal ganglia–thalamo–cortical coordination in humans.

**Contributions:**

1. **Model:** The first deep nonlinear framework that explicitly disentangles shared vs. private subspaces in multi-region intracranial recordings, with new alignment and disentanglement losses tailored for LFP.

2. **Application:** The first demonstration of disentangled latent modeling in pediatric DBS recordings, revealing stimulation-specific reorganization of shared dynamics.

**Code availability.** All code, configs, and scripts to reproduce results are available at `https://github.com/SPIRE-Anonym/spire-ICLR2026`.

## 2 RELATED WORK

**Early latent subspace models.** Classical methods such as Gaussian Process Factor Analysis (GPFA) (Yu et al., 2008) and Canonical Correlation Analysis (CCA) (Bach & Jordan, 2005) extract low-dimensional neural representations but assume linearity and do not distinguish shared from private factors. Semedo et al. (Semedo et al., 2014) extended CCA to identify shared subspaces across populations, laying early foundations for shared/private decomposition. DLAG (Delayed Latents Across Groups) (Gokcen et al., 2022) further partitioned shared and private processes within a probabilistic state-space framework, but under linear–Gaussian assumptions and primarily applied to spiking data. Spectral and multi-population extensions (Gokcen et al., 2023; Oganesian et al., 2024) assume that shared latents dominate inter-regional information, but their applicability to externally perturbed dynamics such as DBS remains unclear.

**Shared/private disentanglement across modalities.** Disentanglement of shared and private factors has been studied in multimodal representation learning. SharedAE (Yi et al.) and multimodal VAEs such as MMVAE (Shi et al., 2019) and DMVAE (Lee & Pavlovic, 2021) align shared subspaces while preserving modality-specific components. While conceptually related to our approach, these methods are designed for neural–behavioral or cross-modal datasets rather than multi-region intracranial recordings under stimulation. Other deep generative models such as LFADS (Pandarinath et al., 2018) uncover latent neural dynamics but treat all latent dimensions as a unified space, without explicit separation of shared and private components.

**Nonlinear multi-region frameworks.** Recent directions emphasize manifold geometry or inter-regional communication. MARBLE (Gosztolai et al., 2025) uses geometric deep learning to uncover interpretable latent manifolds, while MRDS-IR (Dowling & Savin) combines nonlinear local dynamics with linear impulse-response channels to capture directed connectivity. Other work has modeled multi-region LFPs with factor analysis or dynamical systems approaches (Ulrich et al., 2014; Bong et al., 2020; Yang et al., 2021; Schmidt et al., 2020), but typically treats the network as a unified latent space. By contrast, **SPIRE** provides a deep nonlinear framework that explicitly disentangles shared and private subspaces, making it uniquely suited to reveal how DBS reorganizes network-level dynamics in human intracranial recordings.

## 3 SPIRE FRAMEWORK

### 3.1 MODEL OVERVIEW

We introduce **SPIRE** (Shared–Private Inter-Regional Encoder), a dual-latent autoencoder that disentangles shared (cross-regional) and private (region-specific) neural dynamics in multi-region recordings. Each region $r \in R$ is assigned its own encoder–decoder pair.

**Encoders.** Given multichannel inputs $x^{(r)} \in \mathbb{R}^{B \times T \times C_r}$ (batch $B$, length $T$, $C_r$ channels), a Gated Recurrent Unit (GRU) encoder (Cho et al., 2014) produces hidden states $h^{(r)}$, which are linearly projected into shared $z_{\text{sh}}^{(r)}$ and private latent $z_{\text{pr}}^{(r)}$ sequences ($d_{\text{sh}}, d_{\text{pr}}$ are shared and private latent dimensionalities):

$$z_{\text{sh}}^{(r)} = W_{\text{sh}}^{(r)} h^{(r)} \in \mathbb{R}^{B \times T \times d_{\text{sh}}}, \qquad z_{\text{pr}}^{(r)} = W_{\text{pr}}^{(r)} h^{(r)} \in \mathbb{R}^{B \times T \times d_{\text{pr}}}, \tag{1}$$

**Decoders.** Each region has a decoder $f_{\text{dec}}^{(r)}$ that reconstructs its input from the concatenated latents:

$$\hat{x}^{(r)} = f_{\text{dec}}^{(r)}\left(\left[z_{\text{sh}}^{(r)}, z_{\text{pr}}^{(r)}\right]\right) \in \mathbb{R}^{B \times T \times C_r}. \tag{2}$$

**Cross-regional alignment.** To compare and transfer shared dynamics across regions, SPIRE aligns and maps shared latents between pairs:

$$\tilde{z}_{\text{sh}}^{(s \to r)} = M^{(s \to r)} \text{ConvAlign}\left(z_{\text{sh}}^{(s)}\right), \tag{3}$$

where $M^{(s \to r)} \in \mathbb{R}^{d_{\text{sh}} \times d_{\text{sh}}}$ is a lightweight linear mapper initialized to the identity, and ConvAlign is a depthwise 1D convolution over time with an odd kernel size $(2K+1)$ initialized as an impulse. ConvAlign provides lightweight temporal alignment across regions—tolerating small mis-phasings while remaining near identity via regularization—and, because mappings are directional $(s \to r)$ vs $(r \to s)$, the learned alignments need not be symmetric. Concretely, ConvAlign maintains one filter per shared latent dimension; stacking them yields $\mathbf{K}_{s \to r} \in \mathbb{R}^{d_{\text{sh}} \times (2K+1)}$. These modules are regularized during training (Section 3.2) to remain close to identity while allowing flexible temporal alignment and subspace rotation.

## 3.2 Training objective

SPIRE is optimized with a multi-objective loss that balances three goals: (i) faithful reconstruction of each region's activity, (ii) alignment of shared latents across regions, and (iii) disentanglement of shared from private components (Schematic shown in Figure 1).

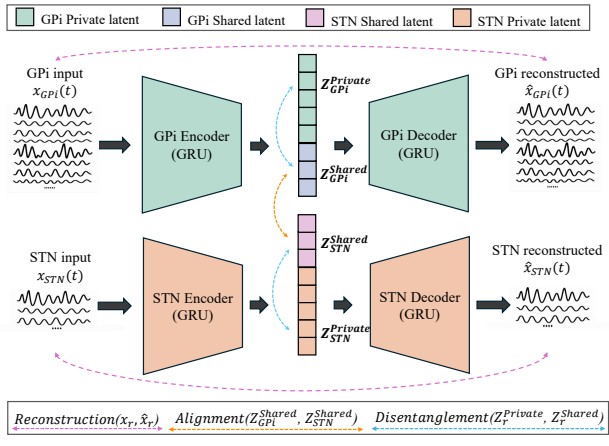

Figure 1: Schematic of model with main goals

**Total loss.** The training objective is a weighted sum:

$$\begin{aligned}
\mathcal{L} = {} & \lambda_{\text{rec}}\mathcal{L}_{\text{rec}} + \lambda_{\text{cross}}\mathcal{L}_{\text{cross}} + \lambda_{\text{self}}\mathcal{L}_{\text{self}} + \lambda_{\text{align}}\mathcal{L}_{\text{align}} + \lambda_{\text{orth}}\mathcal{L}_{\text{orth}} \\
& + \lambda_{\text{mapid}}\mathcal{L}_{\text{mapid}} + \lambda_{\text{align-reg}}\mathcal{L}_{\text{align-reg}} + \lambda_{\text{var-sh}}\mathcal{L}_{\text{var-sh}} + \lambda_{\text{var-pr}}\mathcal{L}_{\text{var-pr}}.
\end{aligned} \tag{4}$$

Dataset-specific choices for $\{\lambda_i\}$ and their schedules are reported in Appendices A.2.2 and A.3.5.

**Reconstruction losses.** $\mathcal{L}_{\text{rec}}$ ensures each region can be reconstructed from its own shared+private latents. To encourage shared latents to carry meaningful variance, we also include two auxiliaries: *cross-reconstruction* ($\mathcal{L}_{\text{cross}}$) which uses only another region's shared latents, and *self-reconstruction* ($\mathcal{L}_{\text{self}}$) which uses only a region's own shared latents.

**Alignment.** $\mathcal{L}_{\text{align}}$ applies variance–invariance–covariance regularization (VICReg, Bardes et al., 2022) between shared latents from different regions, after temporal alignment modules. This enforces overlap between shared subspaces while allowing region-specific views.

**Disentanglement.** Two terms promote separation within each region. First, $\mathcal{L}_{\text{orth}}$ penalizes the Frobenius norm of the cross-covariance between per-feature standardized shared and private latents, reducing redundancy. Second, the optional variance guards ($\mathcal{L}_{\text{var-sh}}$, $\mathcal{L}_{\text{var-pr}}$) prevent degenerate solutions by nudging the standard deviation of shared latents toward 1 and enforcing a minimum variance floor on private latents.

**Regularization of alignment modules.** We use two alignment-specific penalties: $\mathcal{L}_{\text{mapid}}$, which softly biases the linear mappers $M^{(s \to r)}$ toward identity, and $\mathcal{L}_{\text{align-reg}}$, which regularizes the ConvAlign filters toward impulse-like, unit-sum kernels. These terms keep learned alignments close to identity (for interpretability) while still permitting flexible temporal adjustments.

Full mathematical definitions are given in Appendix A.1.1. Moreover, an ablation analysis has been performed and results are reported in Appendix A.2.4

### 3.3 IMPLEMENTATION DETAILS

All models are implemented in PyTorch with GRU encoders–decoders and trained using Adam optimization (Kingma & Ba, 2017) with early stopping. We use standard stability practices (gradient clipping, mixed precision, learning-rate scheduling). Further implementation details, including lag augmentation are provided in Appendix A.1.2.

## 4 SYNTHETIC DATA VALIDATION

### 4.1 DATASET

To validate SPIRE against ground truth, we designed three synthetic datasets with explicitly defined shared and private latent sources, inspired by the statistical structure of intracranial LFPs. Each dataset contained 100 trials of $T = 250$ samples (0.5s at 500Hz), with three shared and three private latent dimensions per region. We use three presets: **D0** linear mixing with Gaussian noise; **D1** adds region-mismatched nonlinear warps and bilinear mixing with $1/f$ noise and AR(1) latents; **D2** extends D1 with a slow time-varying inter-regional lag (sinusoidal) to model phase-dependent communication and latency variability (Bedard et al., 2006; He, 2014; Dupré la Tour et al., 2017; Fries, 2005; 2015; Lakatos et al., 2008).

These regimes progressively increase complexity, from linear settings to nonlinear distortions and temporal misalignments. Further generator details are provided in Appendix A.2.1 and examples of observed channels and latents are illustrated in Figure A.1.

### 4.2 RESULTS

**Qualitative comparison.** To illustrate model performance, we selected dataset D1 (region-mismatched warp with nonlinear mixing). We trained SPIRE and DLAG (fitting details in Appendix A.2.3) separately, extracted shared and private latents, and applied CCA to align each model's shared latents with the ground truth. For SPIRE, we additionally incorporated lag-augmented input features (Eq. A.8), which improves robustness to temporal misalignment; DLAG was run in its standard form without lags. Figure 2 shows the mean $\pm$ SEM across trials for the three shared dimensions of shared latents of region 1. SPIRE's aligned latents track the ground-truth trajectories more closely, yielding higher CCA correlations (0.92, 0.91, 0.71) compared to DLAG (0.86, 0.79, 0.60). This demonstrates that SPIRE recovers shared structure reliably even in the presence of nonlinear distortions, where DLAG begins to degrade.

**Quantitative evaluation.** We further evaluated the performance of SPIRE model and DLAG by training them with four random seeds on all three datasets (D0–D2), and we computed CCA correlations of recovered latents with ground-truth shared and private latents. Results were averaged across both regions (shared1/2 or private1/2) and evaluated statistically using linear models.

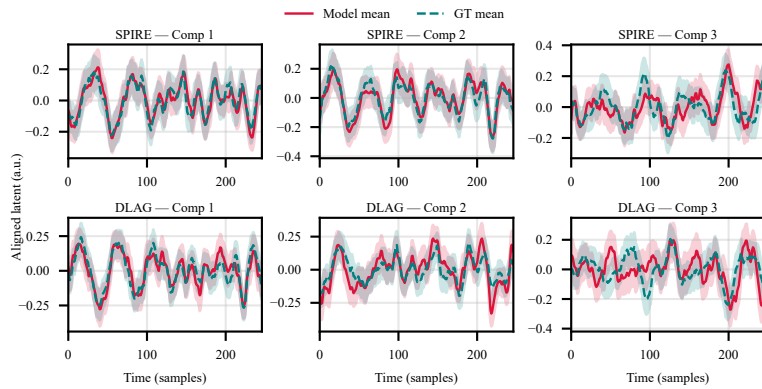

Figure 2: Visual comparison of shared latents of region 1 in the nonlinear regime (D1). SPIRE (left panels) and DLAG (right panels) are CCA-aligned to the ground truth. Solid lines show trial-averaged latent trajectories, with shaded areas denoting $\pm$ SEM across trials. SPIRE achieves consistently higher alignment with ground-truth shared latents. *Both models were evaluated on the identical dataset; apparent GT differences reflect that CCA alignment was performed separately for each model.*

Figure 3 summarize these results. SPIRE is statistically significantly better than DLAG in retrieving ground-truth private latents, and also outperforms DLAG on shared latents in the nonlinear (D1) and time-varying delay (D2) regimes, though not with statistical significance. We note that D0 is a linear, DLAG-friendly regime and less representative of real neural data; our conclusions therefore emphasize the more realistic D1 and D2 settings.

Together, these results demonstrate that SPIRE reliably disentangles shared and private structure under conditions that approximate real intracranial data.

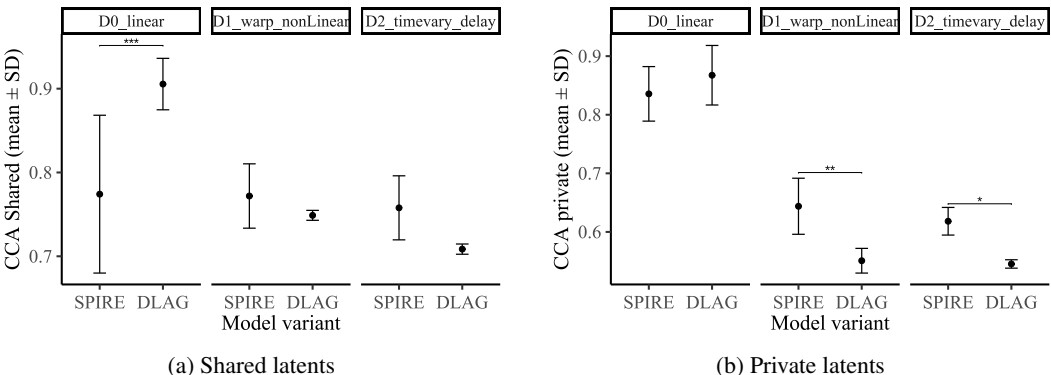

(a) Shared latents                      (b) Private latents

Figure 3: Statistical comparison of DLAG and SPIRE. Mean $\pm$ SD CCA across four seeds, faceted by dataset regime. SPIRE outperforms DLAG in nonlinear (D1) and time-varying delay (D2) regimes for shared latents, and is statistically significantly better in retrieving private latents. Significance: $* : p < 0.05$, $** : p < 0.01$, $*** : p < 0.001$.

## 5 APPLICATION TO HUMAN DBS RECORDINGS

### 5.1 DATASET

We analyzed intracranial local field potentials (LFPs) from ten pediatric patients (ages 5–23) with clinically diagnosed dystonia who underwent stereotactic implantation of DBS leads (Anonymous,

2019; 2024a; 2018). Electrodes analyzed in this study, targeted the globus pallidus internus (GPi) and subthalamic nucleus (STN). Recordings were obtained both during stimulation-off and during stimulation at clinically relevant frequencies (GPi stimulation: 85, 185, 250 Hz and STN stimulation: 85, 185 Hz). Analyses were performed at the hemisphere level, as most patients were implanted bilaterally.

Signals were originally sampled at 24,414 Hz and re-referenced using bipolar derivations to suppress common-mode noise (Mercier et al., 2022). To reduce dimensionality while retaining relevant neural dynamics, data were downsampled to 500 Hz, notch filtered and band-limited with a 50 Hz low-pass Butterworth filter. This preprocessing removed high-frequency stimulation artifacts (Alarie et al., 2022) while preserving oscillatory components of interest. Each hemisphere's recordings were segmented into non-overlapping 0.5 s windows. To capture short-timescale temporal dependencies, we augmented each channel with time-lagged versions (0–3 lags), effectively expanding the input feature space(Appendix A.8). Additional details on participant demographics, stimulation paradigms, and preprocessing are provided in Appendix A.3.1, A.3.2 and A.3.3.

## 5.2 TRAINING

Because of variability across subjects in disease etiology, anatomy, demographics, and number of available recording contacts, we trained SPIRE across a grid of shared (3–5) and private (2–4) latent dimensions. We selected the best configuration for each case based on variance of each latent type and total validation loss. The procedure for model selection is explained in details in Appendix A.3.4. Figure A.3 underscores that SPIRE does not impose a fixed decomposition but adapts to the statistical structure present in each subject's recordings.

## 5.3 RESULTS

### 5.3.1 DISENTANGLING SHARED AND PRIVATE LATENT DYNAMICS

Figure 4 shows two examples of representative Uniform Manifold Approximation and Projection (UMAP) (McInnes et al., 2018) and time-domain presentations of extracted latents of test set. In S3_R (balanced), 3D UMAPs show substantial overlap between shared GPi and STN latents while private and shared clusters of same region remain distinct; time traces confirm that paired shared dimensions co-fluctuate with small lags, whereas private dimensions diverge with region-specific transients. In S8_R (private-dominant), shared GPi/STN traces are still aligned in phase but exhibit very low-amplitude, slow baseline trends (global co-modulation) with little high-variance structure; meanwhile, private GPi and STN latents show larger, region-specific dynamics in time. Together, these examples illustrate that SPIRE captures genuine cross-regional coupling when present (S3_R) and otherwise allocates most variance to private latents while the shared latents encode only weak, slow global fluctuations (S8_R).

We quantified (Figure 5a) subspace similarity with top-k CCA among all subject_hemispheres. After alignment, the shared GPi/STN spaces showed near-unit canonical correlation (median around 1.0), indicating a common manifold across regions. In contrast, CCA between shared and private subspaces within each region was markedly lower (medians 0.55–0.65), consistent with successful disentanglement: private latents carry region-specific variance while shared latents encode cross-regional structure. Residual shared–private correlation likely reflects low-frequency global trends that are not perfectly orthogonal by design.

To further validate the decomposition, we evaluated reconstruction accuracy on held-out test data using different subsets of latents (Figure 5b). Reconstructions from the full latent space (shared + private) achieved near-zero error (medians: 0.00211 and 0.000983, for GPi and STN respectively), confirming the model's capacity to faithfully recover observed signals. When restricted to private latents, reconstruction error increased (medians: 0.544 and 0.391, for GPi and STN respectively), indicating that private subspaces alone do not contain enough information to explain the data. In contrast, same-region shared latents supported substantially better reconstructions, demonstrating that the bulk of recoverable neural dynamics lies in the shared manifold between GPi and STN (medians: 0.0462 and 0.0178, for GPi and STN respectively). Notably, cross-region shared latents (e.g., using STN shared latents to reconstruct GPi) performed worse than same-region shared latents, reflecting that while the shared space encodes overlapping dynamics, each region's decoder relies

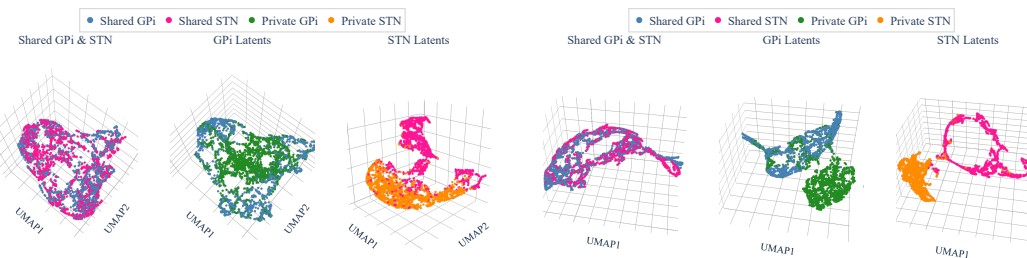

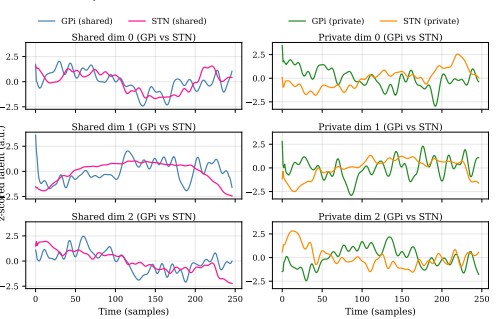
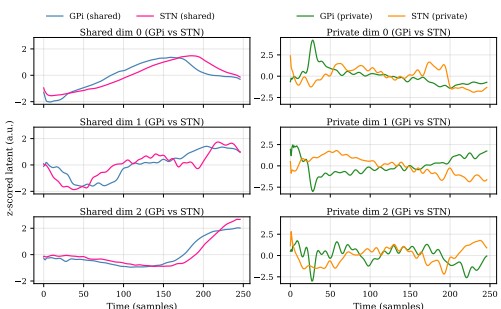

(a) S3_R — UMAP of shared vs. private latents (balanced case).

(b) S8_R — UMAP of shared vs. private latents (private-dominant case).

(c) S3_R — time-domain latents (shared aligned; private higher variance).

(d) S8_R — time-domain latents (shared low-amplitude with lag; private higher variance).

Figure 4: Shared–private structure differs across subjects. Panels (a,b) show 3D UMAPs: in S3_R, shared GPi/STN embeddings largely overlap while shared and private clusters of both regions remain distinct; in S8_R, private vs shared of both regions separate strongly and shared embeddings intermix less. Panels (c,d) plot paired latent traces for three dims: S3_R exhibits well-aligned shared trajectories (small lag) with clearly distinct private trajectories. In S8_R, shared GPi/STN trajectories are still *phase-aligned* but have *very low amplitude* and primarily reflect *slow baseline co-modulation*, whereas private trajectories carry larger, region-specific dynamics that dominate the variance. These examples complement the cohort-level variance partition (Fig. A.3).

on a region-specific view of this manifold. Together, these results show that SPIRE captures a dominant shared structure driving both regions, with private latents encoding complementary but non-redundant residual variance (variance analysis detailed in Appendix A.3.4).

**Comparison to baselines.** To evaluate how SPIRE compares to existing multi-view latent factorization methods, we benchmarked its reconstruction performance against two recent baselines: Shared Autoencoder (SharedAE) (Yi et al.) and Multi-modal VAE (MMVAE) (Shi et al., 2019). We also attempted to apply DLAG (Gokcen et al., 2022), however, despite extensive efforts, DLAG consistently failed to converge on our intracranial recordings, yielding numerical instabilities during Gaussian process optimization. Details of the models, our implementation and adaptation of SharedAE and MMVAE and DLAG's failure modes are provided in Appendix A.3.6. Because the released SharedAE implementation produces one embedding vector per window (rather than time-resolved latent sequences), we restrict its evaluation to reconstruction and do not report trajectory-level analyses such as per-timepoint CCA or decoding (Appendix A.3.6). As shown in Figure 6, SPIRE consistently achieves lower MSE across both GPi and STN regions. Notably, MMVAE lacks architectural disentanglement of shared and private latents, and therefore only supports full reconstructions. In contrast, SPIRE's explicit separation enables partial reconstructions that isolate private and shared contributions.

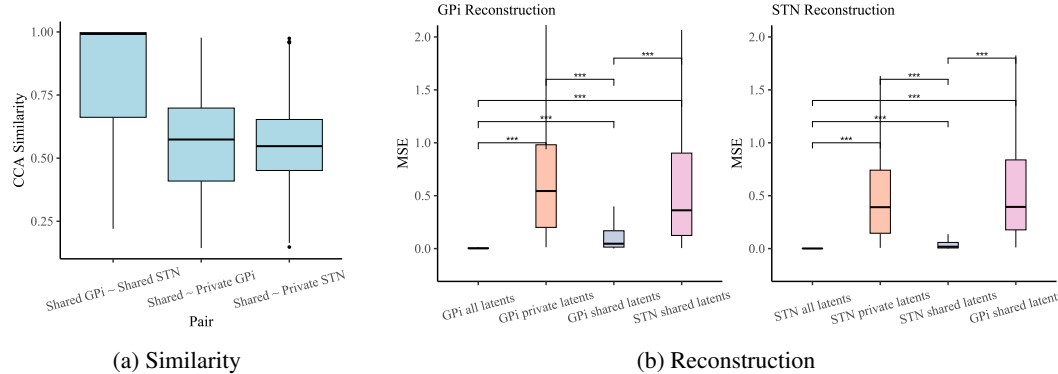

(a) Similarity        (b) Reconstruction

Figure 5: Validation of shared–private separation. (a) CCA between latent subspaces shows that shared GPi and STN latents are highly correlated with each other, while their leakage into private latents is weak—supporting effective disentanglement. (b) Reconstruction accuracy on held-out test data indicates that full latents (shared+private) yield the lowest error; private-only performs worst; while shared latents enable substantially better reconstructions, with same-region shared outperforming cross-region shared. Together, these analyses confirm that shared latents encode the dominant cross-regional structure, while private latents capture residual, region-specific variance. Boxplots: median, IQR, whiskers $1.5\times$IQR. Statistics: linear mixed-effects with Tukey correction ($n = 17$) (Tukey, 1949). Significance: $*: p < 0.05, **: p < 0.01, ***: p < 0.001$.

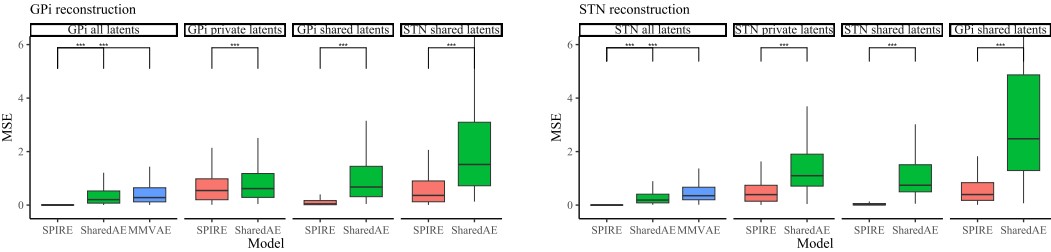

Figure 6: Reconstruction performance of SPIRE compared to SharedAE and MMVAE across GPi and STN regions. SPIRE achieves lower reconstruction error across all latent conditions, highlighting its advantage in modeling region-specific and shared dynamics. Boxplots indicate the median and interquartile range, with whiskers extending to 1.5×IQR. Significance: $*: p < 0.05$, $**: p < 0.01, ***: p < 0.001$.

### 5.3.2 SHARED LATENTS ENCODE STIMULATION FREQUENCY

We analysed whether latent representations encode stimulation-specific information. Random Forest (Breiman, 2001) classifiers were trained to decode stimulation condition from each latent type, using raw timepoint-level embeddings without temporal averaging (80/20 split, standardized, 100 trees). For GPi stimulation, classifiers decoded four conditions (Off, 85, 185, 250 Hz; $n = 17$ hemispheres), and for STN stimulation three conditions (Off, 85, 185 Hz; $n = 13$). As shown in Figure 7, both private and shared latents supported above-chance classification of stimulation frequency, demonstrating that stimulation consistently perturbs the latent activity space. Crucially, shared latents yielded significantly higher decoding accuracy than private latents in both datasets (linear mixed-effects model with Tukey correction, $p < 0.001$), with no difference between GPi- and STN-derived shared spaces. These results suggest that shared latents encode robust stimulation-related signatures that generalize across regions, while private latents capture more variable, region-specific responses.

For completeness, we also quantified distributional shifts using Maximum Mean Discrepancy (MMD) (Gretton et al., 2012), which confirmed frequency-dependent divergence in both shared and private latents (Appendix A.3.7).

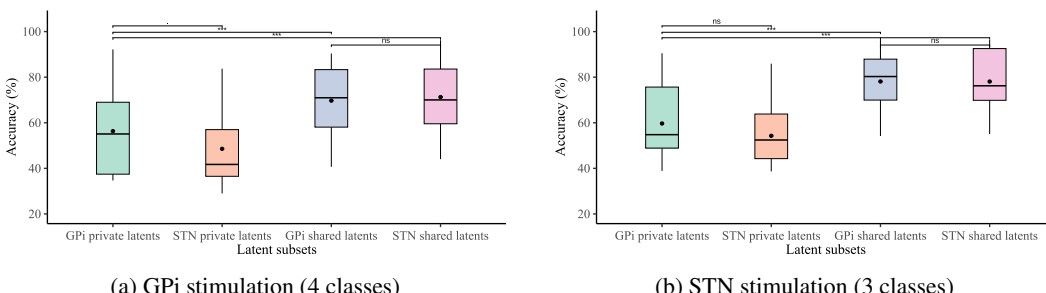

(a) GPi stimulation (4 classes)   (b) STN stimulation (3 classes)

Figure 7: Shared latents decode stimulation condition better than private latents across stimulation sites. Boxplots show Random Forest test accuracy for each latent subset; black dots denote means, boxes show medians and IQRs, whiskers extend to $1.5 \times$IQR. Statistics from linear mixed-effects models with Tukey-corrected pairwise comparisons; significance codes: $.:p < 0.1$, $*:p < 0.05$, $**:p < 0.01$, $***:p < 0.001$, "ns" = not significant. Within each dataset, GPi shared and STN shared accuracies are not significantly different, while both shared subsets outperform the private subsets ($p < 0.001$).

## 6 DISCUSSION

Our results demonstrate that deep shared–private factorization provides a powerful lens for understanding how external perturbations reorganize multi-region neural dynamics. By training SPIRE on baseline recordings, we established a reference frame of intrinsic coordination and then showed that stimulation selectively reorganizes the shared latent space. This highlights shared latents as a compact, stimulation-sensitive substrate of cross-regional dynamics, while private latents capture residual, local activity.

**Neuroscience insights.**   In pediatric dystonia data, shared latents consistently encoded stimulation-specific signatures that generalized across sites and frequencies. This supports circuit-level theories of DBS that emphasize distributed network modulation rather than isolated local effects (McIntyre & Hahn, 2010; Yang et al., 2021; Schmidt et al., 2020). The fact that stimulation frequency could be decoded directly from shared latents suggests that DBS reorganizes coordination patterns in a systematic, frequency-dependent manner.

**Methodological significance.**   Our findings illustrate how training on baseline dynamics can be leveraged to quantify reorganization under perturbation, a paradigm broadly applicable to multi-view dynamical systems. Methodologically, SPIRE complements existing latent models such as DLAG and multimodal VAEs by providing a nonlinear yet simple architecture with explicit factorization. Its reliance on a small set of interpretable losses (reconstruction, alignment, disentanglement) makes the method easy to reproduce and adapt—an important strength in neuroengineering where reproducibility is often a barrier.

**Limitations and outlook.**   The present work is restricted to relatively short-timescale stimulation and to LFP signals alone. Future work will extend SPIRE to longer stimulation paradigms, integrate spiking and field potentials, and incorporate probabilistic objectives for uncertainty quantification. Because SPIRE can naturally extend to more than two regions, assessing generalization to additional regions (e.g., cortex and thalamus), etiologies, and chronic timescales is a natural next step. Although stimulation-on data can in principle contain artifacts, our preprocessing pipeline (bipolar rereferencing and 50 Hz low-pass with stimulation fundamentals $\geq 85$ Hz) substantially attenuates artifact energy, and SPIRE itself is trained only on off-stimulation data. This reduces the risk that latent structure reflects artifacts rather than physiology, though residual low-frequency consequences of stimulation cannot be fully excluded. Finally, SPIRE's latents should be viewed as statistical abstractions: assigning precise biophysical meaning to individual dimensions will require complementary experiments and multimodal validation.

ETHICS STATEMENT

All human neural data analyzed in this study were collected under protocols approved by the Institutional Review Boards (IRBs) of the participating medical centers. Written informed consent was obtained from all participants or their legal guardians (for minors), with assent obtained from children when appropriate. Data handling complied with HIPAA and institutional privacy policies: recordings were de-identified prior to analysis, stored on secure, access-controlled systems, and used solely for research purposes. This work does not involve clinical decision-making, patient intervention beyond standard clinical care, or vulnerable-population recruitment beyond the clinical cohort already undergoing DBS evaluation. No attempt was made to re-identify participants. Due to ethical and regulatory constraints governing research with human subjects, the raw neural recordings cannot be publicly released; de-identified data may be shared upon reasonable request and execution of a data use agreement consistent with IRB approvals. The authors affirm compliance with the ICLR Code of Ethics and disclose no conflicts of interest or external sponsorship that could influence the work.

REPRODUCIBILITY STATEMENT

All code used in this study is released to support reproducibility at `https://github.com/SPIRE-Anonym/spire-ICLR2026`. Complete explanation of data preprocessing and training details are provided in Sections 3.2, 5.1 and Appendices A.2.1, A.2.2, A.3.3, A.3.5.

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

# A  APPENDIX

## A.1  SPIRE DETAILS

### A.1.1  SPIRE LOSS TERMS

Let $\mathcal{R}$ denote the set of regions. For any region $r \in \mathcal{R}$, let $x^{(r)} \in \mathbb{R}^{B \times T \times C_r}$ be the input and $\hat{x}^{(r)}$ its reconstruction. Let $z_{\text{sh}}^{(r)} \in \mathbb{R}^{B \times T \times d_{\text{sh}}}$ and $z_{\text{pr}}^{(r)} \in \mathbb{R}^{B \times T \times d_{\text{pr}}}$ be shared and private latents, and $\tilde{z}_{\text{sh}}^{(s \to r)} = M^{(s \to r)} \text{ConvAlign}(z_{\text{sh}}^{(s)})$ the temporally aligned and linearly mapped shared latents from $s$ into the space of $r$ (see main text). When needed, we flatten time and batch so that $Z_{\bullet}^{(r)} \in \mathbb{R}^{(BT) \times d_{\bullet}}$, write $N = BT$, and use

$$\text{Cov}(A, B) = \tfrac{1}{N-1} A^{\top} B, \qquad \text{std}(\cdot) \text{ computed per feature over the } N \text{ rows.}$$

**Reconstruction.**

$$\mathcal{L}_{\text{rec}} = \sum_{r \in \mathcal{R}} \left\| x^{(r)} - \hat{x}^{(r)} \right\|_2^2. \tag{A.1}$$

**Cross- and self-reconstruction.**  (ordered pairs; two directions in the two-region case)

$$\mathcal{L}_{\text{cross}} = \sum_{\substack{r,s \in \mathcal{R} \\ r \neq s}} \left\| x^{(r)} - f_{\text{dec}}^{(r)}(\tilde{z}_{\text{sh}}^{(s \to r)}, 0) \right\|_2^2, \qquad \mathcal{L}_{\text{self}} = \sum_{r \in \mathcal{R}} \left\| x^{(r)} - f_{\text{dec}}^{(r)}(z_{\text{sh}}^{(r)}, 0) \right\|_2^2. \tag{A.2}$$

**Alignment (VICReg).**  Computed after alignment/mapping and flattening (as in VICReg):

$$\mathcal{L}_{\text{align}} = \tfrac{1}{2} \sum_{\substack{r,s \in \mathcal{R} \\ r \neq s}} \left( \text{VICReg}(z_{\text{sh}}^{(r)}, \tilde{z}_{\text{sh}}^{(s \to r)}) + \text{VICReg}(z_{\text{sh}}^{(s)}, \tilde{z}_{\text{sh}}^{(r \to s)}) \right). \tag{A.3}$$

**Orthogonality.**  Per-feature standardization precedes the cross-covariance penalty:

$$\mathcal{L}_{\text{orth}} = \sum_{r \in \mathcal{R}} \left\| \text{Cov}(\bar{Z}_{\text{sh}}^{(r)}, \bar{Z}_{\text{pr}}^{(r)}) \right\|_F^2, \tag{A.4}$$

where $\bar{Z}$ indicates zero-mean, unit-variance standardization per feature.

**Variance guards.**

$$\mathcal{L}_{\text{var-sh}} = \sum_{r \in \mathcal{R}} \left\| \text{std}(z_{\text{sh}}^{(r)}) - 1 \right\|_2^2, \qquad \mathcal{L}_{\text{var-pr}} = \sum_{r \in \mathcal{R}} \sum_{j=1}^{d_{\text{pr}}} \max\!\left(0, \, \tau - \text{std}(z_{\text{pr},j}^{(r)})\right)^2, \qquad \text{(A.5)}$$

with target $\tau > 0$.

**Alignment-module regularizers.** Linear mappers are biased toward identity:

$$\mathcal{L}_{\text{mapid}} = \sum_{s \to r} \left\| M^{(s \to r)} - I \right\|_F^2, \qquad \text{(A.6)}$$

and ConvAlign filters toward impulse-like, unit-sum kernels. With per-dimension kernels $\mathbf{k}_{s \to r, j} \in \mathbb{R}^{2K+1}$ and centered impulse $\delta \in \mathbb{R}^{2K+1}$,

$$\mathcal{L}_{\text{align-reg}} = \sum_{s \to r} \sum_{j=1}^{d_{\text{sh}}} \left( \left\| \mathbf{k}_{s \to r, j} - \delta \right\|_2^2 + \left\| \mathbf{1}^\top \mathbf{k}_{s \to r, j} - 1 \right\|_2^2 \right). \qquad \text{(A.7)}$$

### A.1.2 IMPLEMENTATION DETAILS

**Lag augmentation.** To improve robustness to temporal misalignment, SPIRE receives lag-augmented features. For region 1 (GPi) with $C$ channels and sequence length $T$, we stack the signal and its lagged versions up to lag $L$:

$$X^{(1)} = \begin{bmatrix} x_{:,\, 0:T-L}^{(1)} \\ x_{:,\, 1:T-L+1}^{(1)} \\ \vdots \\ x_{:,\, L:T}^{(1)} \end{bmatrix} \in \mathbb{R}^{C(L+1) \times (T-L)}, \qquad \text{(A.8)}$$

and analogously for region 2 (STN). This expands each input into $(L+1)$ lagged copies, enabling SPIRE to model short-range delays directly.

**Training setup.** Data are split 80/20 into training and validation sets, with mini-batches of size 8 for training and batch size 1 for validation. Learning rate is initialized at $10^{-3}$ and halved on plateaus (patience 10 epochs). Gradients are scaled and clipped to an L2 norm of 1.0. Validation loss is monitored each epoch with early stopping (patience 20, ignoring the first 140 epochs).

## A.2 SYNTHETIC DATA

### A.2.1 SYNTHETIC DATA GENERATOR

Our synthetic generator builds multichannel observations from ground-truth shared and private latent processes.

**Latent dynamics.** Shared and private latents are generated as bursty oscillators with variable base frequencies, occasional higher-frequency bursts (30–40 Hz), and occasional frequency jumps. Optional AR(1) filtering introduces temporal autocorrelation. Region-specific monotone warps (region_mismatch) or cubic distortions may be applied, and the second region may receive time-varying delayed copies (sinusoidal lag profile).

**Observation mixing.** Each region has an electrode geometry with contacts arranged in rows. Latents are linearly projected to contacts through randomized spatial mixing kernels, followed by optional nonlinear mixing (gain, bilinear, or both). Cross-terms between shared and private sources may also be added.

**Noise structure.** We incorporate multiple biologically inspired noise sources: (i) $1/f$ colored noise added to latents, (ii) heteroscedastic noise where variance scales with latent amplitude (abs/power/multiplicative modes), (iii) slow row-common drifts across contacts, and (iv) low-rank common-mode fluctuations across all contacts. Finally, bipolar re-referencing is applied to suppress global noise.

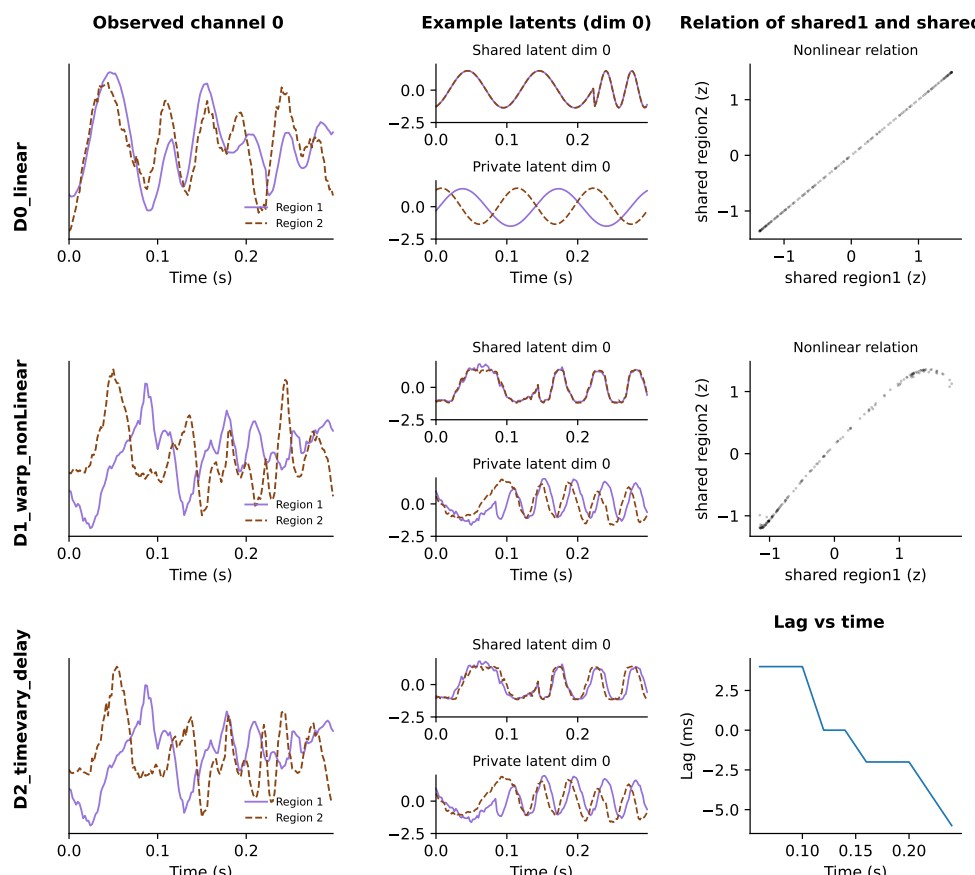

Figure A.1: Synthetic datasets. The trials are 0.5 s but for visualization purposes we have shown only 0.3 s. left: example observed channels (region 1 solid purple, region 2 dashed brown). middle: example shared (top) and private (bottom) latents (region 1 solid purple, region 2 dashed brown). right: diagnostic plots showing nonlinear warping (top two rows) and time-varying delays (bottom row) between shared latent dim 0 of two regions.

**Datasets.** The three benchmark presets used in this work were:

- `D0_linear`: linear mapping, no nonlinearities, low Gaussian noise.
- `D1_warp_nonLinear`: nonlinear mixing (gain+bilinear), region-mismatched warp, $1/f$ noise, AR(1).
- `D2_timevary_delay`: same as above with sinusoidal time-varying lag of amplitude 3 samples.

Each dataset contains 100 trials of 250 timepoints at 500 Hz, with 3 shared and 3 private latent dimensions per region.

Figure A.1 illustrates example observed channels, ground-truth shared and private latents, and a diagnostic plot of sharedness for the three regimes (D0–D2).

### A.2.2 TRAINING DETAILS

**Model architecture.** Each region used a GRU encoder/decoder (hidden size 64, dropout 0.3). Encoders projected into shared ($d_s = 3$) and private ($d_p = 3$) latents. Cross-regional alignment was performed with depthwise ConvAlign filters (kernel size 9) followed by lightweight linear mappers initialized as identity. ConvAlign filters were regularized toward impulse responses.

Table A.1: Loss weights for synthetic training across phases. $\alpha_p$ ramps from 0 to 1 between epochs 80–140.

| Loss component | Pre (0–80) | Ramp (80–140) | Post (140–500) |
|---|---|---|---|
| $\mathcal{L}_{\text{rec}}$ (reconstruction) | 1.00 | 1.00 | 1.00 |
| $\mathcal{L}_{\text{align}}$ (shared alignment) | 0.22 | 0.10 | 0.08 |
| $\mathcal{L}_{\text{cross}}$ (cross recon) | 0.03 | 0.05 | 0.07 |
| $\mathcal{L}_{\text{self}}$ (self recon) | 0.02 | 0.04 | 0.03 |
| $\mathcal{L}_{\text{orth}}$ (shared$\perp$private) | 0.008 | 0.015 | 0.025 |
| $\mathcal{L}_{\text{mapid}}$ (mapper identity) | 0.010 | 0.005 | 0.000 |
| $\mathcal{L}_{\text{align-reg}}$ (ConvAlign reg.) | $1\times10^{-4}$ | $5\times10^{-4}$ | $1\times10^{-4}$ |
| $\mathcal{L}_{\text{var}}^{\text{shared}}$ (unit var) | 0.005 | 0.005 | 0.005 |
| $\mathcal{L}_{\text{var}}^{\text{private}}$ (floor) | 0.002 | 0.002 | 0.002 |

Freeze window: epochs 90–110 (shared projections frozen, aligners remain trainable).

**Training regime.** Models were trained for up to 500 epochs. A private-gate $\alpha_p$ gradually opened between epochs 80–140, phasing in private latents. During epochs 90–110, shared projections were frozen while aligners remained trainable.

**Loss scheduling.** Loss weights were scheduled in three phases (pre, ramp, post) aligned to the private-gate ramp. Table A.1 summarizes the schedules for all components.

### A.2.3 DLAG FITTING DETAILS

**Data formatting.** Following the public DLAG demo, we represent each trial as a struct with fields `trialId`, `T`, and `y`. For our two regions (GPi-like = group 1; STN-like = group 2), we construct `y` by stacking the trial's multichannel recordings from both regions along the channel dimension, yielding a matrix of shape $[C_1+C_2] \times T$ per trial. A small i.i.d. Gaussian jitter ($\varepsilon \sim \mathcal{N}(0, \sigma^2 I)$, $\sigma \approx 10^{-6}$) is added to avoid singular covariances during initialization. We set the across-group (shared) and within-group (private) latent dimensionalities in DLAG to match the synthetic ground-truth for each preset: 3 shared and 3 private latents.

**Temporal and kernel settings.** We follow the demo configuration and use the RBF Gaussian-process kernel (`covType='rbf'`) for latent dynamics. The GP timescale initialization is `startTau = 2 × binWidth`. The `binWidth` parameter (ms) is set to the sample period used to bin the synthetic trials, and `segLength = T` matches the trial length (in time bins).

**Optimization and stopping.** We fit DLAG with EM up to `maxIters = 500` (demo-style runs may use higher values), log-likelihood tolerance `tolLL = 10⁻⁸`, and progress diagnostics saved every `freqParam = 10` iterations. Private noise floors are enforced with `minVarFrac = 10⁻³`. Unless otherwise noted, we do not perform cross-validation (`numFolds = 0`). We fix a random seed for reproducibility.

**Evaluation protocol.** For each synthetic preset, we train four seeds and align the recovered DLAG latents to ground truth via CCA (performed separately for across-group vs. within-group latents). We then report canonical correlations for shared and private components, averaged across regions, trials, and seeds (Figure 3). This evaluation mirrors the SPIRE protocol: SPIRE's learned shared/private *sequences* are CCA-aligned to ground-truth latent trajectories under identical splitting and statistics.

### A.2.4 ABLATION ANALYSIS

To assess the role of individual loss terms, we constructed ablation variants of SPIRE by setting selected weights to zero (e.g., `abl_no_w_align`, `abl_no_w_orth`), as well as grouped "family" ablations (e.g., removing all alignment-related terms). In addition, we tested four control variants: (i) `ctrl_identity_aligner`, which bypasses convolutional aligners, (ii) `ctrl_no_private_ramp`, which fixes the private scaling to one, (iii) `ctrl_no_freeze`, which

disables the freeze window. (iv) `ctrl_no_var_guard`, which disables the guards on shared and private variances. Each variant was trained with four random seeds on all three datasets (D0–D2), and we computed CCA correlations of recovered latents with ground-truth shared and private latents. Results were averaged across both regions (shared1/2 or private1/2) and evaluated statistically using linear models with Dunnett-adjusted contrasts against the full model (`SPIRE_synth`) as reference.

Figure A.2a and Figure A.2b summarize the ablation analyses. No ablation variant was consistently superior to the full model across all datasets and latent types. The alignment loss had the strongest overall impact: removing it substantially reduced both shared and private recovery. For shared CCA, nearly all loss terms contributed positively across datasets, with performance consistently decreasing when a term was removed. The only exception was the variance guard, where removal led to unstable private recovery, suggesting its role is primarily in stabilizing variance rather than directly boosting mean accuracy. For private CCA, some ablations (e.g., `w_self`, `w_orth`) occasionally produced small improvements or negligible changes, but these same ablations worsened shared recovery. This dissociation indicates that different loss components are critical for different aspects of disentanglement: alignment and disentanglement terms are essential for shared structure, while variance guard and alignment regularization play stronger roles in private stability. Together, these findings confirm that SPIRE's composite loss design is necessary to recover both shared and private dynamics, with complementary terms ensuring robust disentanglement under realistic distortions.

## A.3 REAL INTRACRANIAL DBS RECORDINGS

### A.3.1 PARTICIPANT INFORMATION

Table A.2 summarizes demographics, etiologies, and the number of usable recording and stimulation contacts per hemisphere. Etiologies spanned genetic, metabolic, and acquired forms of dystonia. All subjects were male and between 5 and 23 years of age.

Analyses were performed at the hemisphere level. Based on channel quality thresholds (at least three usable microelectrode contacts per region), 17 hemispheres were included for GPi stimulation experiments and 13 for STN stimulation. Due to clinical time constraints, the 250 Hz condition was not administered in 8 hemispheres with STN stimulation hence we only consider 85 and 185 Hz stimulation for STN stimulation analysis.

### A.3.2 STIMULATION PARADIGM

All procedures were approved by institutional review boards at participating medical centers (Anonymous), and informed consent was obtained in accordance with HIPAA regulations.

Stimulation experiments were conducted 2–3 days post-implantation, while temporary stereoelectroencephalography (sEEG) leads remained in place. Our stimulation protocol (adapted from prior work (Anonymous, 2023)) was designed to probe the frequency-dependent effects of DBS using clinical stimulation ranges commonly used for dystonia. Trains of biphasic pulses were delivered at 85, 185, and 250 Hz through adjacent macro-contacts on each DBS lead. Each frequency–contact pair was tested under two configurations (cathode–anode and reversed polarity), with approximately 1000 pulses delivered per condition (pulse width: 90 μs).

Amplitude was set to 3 V by default but reduced if subjects experienced discomfort. For frequencies above 100 Hz, a ramping protocol gradually increased voltage from 0 to 3 V over 12 seconds to improve tolerance. Intracranial signals from all implanted leads were recorded simultaneously, with recording duration adjusted to ensure that 1000 pulses were captured in each case.

### A.3.3 SIGNAL PRE-PROCESSING (EXTENDED DETAILS)

Recordings were acquired at 24,414 Hz using a Tucker-Davis Technologies (TDT) system and pre-processed initially in MATLAB (R2023b). Bipolar derivation (subtracting adjacent contacts) was applied to enhance spatial specificity and reduce common-mode noise.

After downsampling and filtering (details in Section 5.1), signals were segmented into non-overlapping 0.5-s windows. To capture temporal dependencies, lag-augmented features with 0–3 sample delays were appended to each channel, expanding the feature dimensionality four-fold. This

Table A.2: Participant demographics and number of stimulation and recording contacts per hemisphere. The table includes each subject's etiology, the number of usable microelectrode recording contacts per region (GPi, STN), and whether GPi or STN stimulation was applied in each hemisphere. Abbreviations: MEPAN = Mitochondrial Enoyl CoA Reductase Protein-Associated Neurodegeneration Heimer et al. (2019), KMT2B = Lysine Methyltransferase-2B Abela & Kurian (1993), GA1 = Glutaric aciduria type 1 Boy (2017), PKAN = Pantothenate Kinase-Associated Neurodegeneration Gregory & Hayflick (2017), HIE = Hypoxic-ischemic encephalopathy Vannucci (2000), MYH2 = Myosin heavy chain IIa mutation Tajsharghi et al. (2005), CP = Cerebral palsy Krigger (2006), BG = Basal ganglia.

| Subject | Etiology | Hemisphere | GPi rec. | STN rec. | GPi stim. | STN stim. |
|---------|----------|------------|----------|----------|-----------|-----------|
| S1 | MEPAN | Left | 18 | 3 | 3 | 1 |
|    |       | Right | 12 | 6 | 2 | 1 |
| S2 | GA1 | Left | 6 | 6 | 2 | 1 |
|    |     | Right | 9 | 1 | 2 | 1 |
| S3 | Atypical PKAN | Left | 11 | 4 | 4 | 1 |
|    |               | Right | 8 | 4 | 3 | 1 |
| S4 | HIE | Left | 6 | 3 | 2 | 0 |
|    |     | Right | 3 | 3 | 2 | 0 |
| S5 | CP (Prematurity) | Left | 3 | 1 | 1 | 1 |
|    |                  | Right | 3 | 4 | 1 | 2 |
| S6 | GA1 | Left | 3 | 4 | 2 | 1 |
|    |     | Right | 6 | 0 | 2 | 1 |
| S7 | Kernicterus Hamza (2019) | Left | 9 | 3 | 2 | 1 |
|    |                          | Right | 8 | 4 | 2 | 2 |
| S8 | BG hemorrhage | Left | 6 | 3 | 1 | 0 |
|    |               | Right | 9 | 3 | 1 | 0 |
| S9 | KMT2B | Left | 7 | 4 | 3 | 2 |
|    |       | Right | 7 | 4 | 3 | 2 |
| S10 | HIE | Left | 8 | 4 | 2 | 2 |
|     |     | Right | 8 | 6 | 2 | 1 |

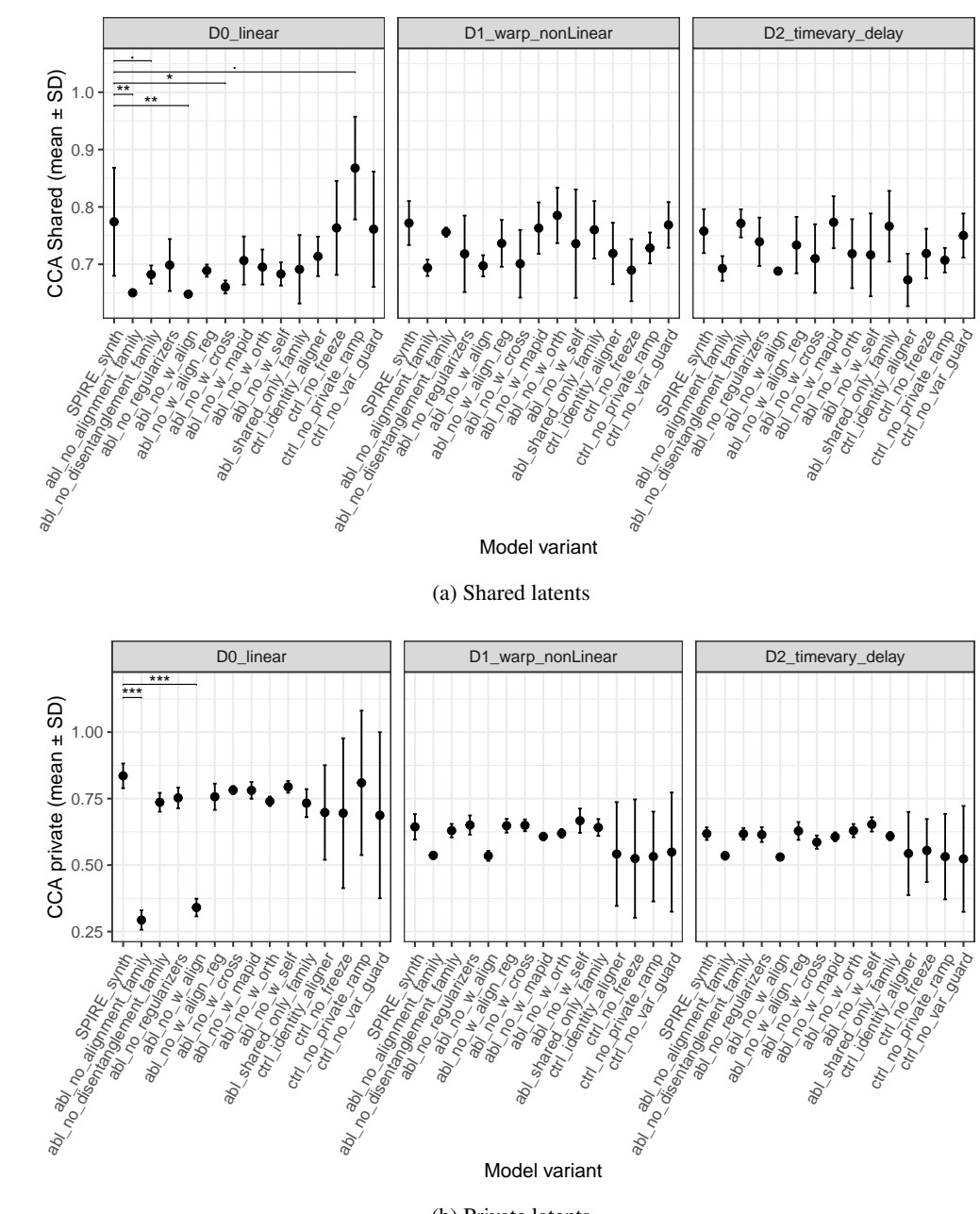

(a) Shared latents

(b) Private latents

Figure A.2: Ablation results for shared latents with statistical comparison. Mean $\pm$ SD CCA across four seeds, faceted by dataset regime. Dunnett-adjusted contrasts are shown relative to the full SPIRE model (`SPIRE_synth`).

choice balanced computational efficiency with the ability to represent short-timescale spatiotemporal dynamics.

Finally, data were split into training (80%) and test (20%) sets for each subject and hemisphere before model fitting.

### A.3.4 VARIANCE PARTITION AND MODEL SELECTION

**Motivation.** Subjects varied widely in etiology, anatomy, and recording coverage, so we allowed latent dimensionalities to adapt per subject–hemisphere. We trained SPIRE across a grid of shared (3–5) and private (2–4) latent dimensions. To compare candidate mode vvls, we quantified how much observed variance in GPi and STN signals was uniquely explained by shared latents, uniquely explained by private latents, or redundantly captured by both. This partitioning ensured that both subspaces contributed meaningfully, avoiding degenerate solutions where all variance collapses into one.

**Fraction of variance explained (FVE).** For each region $r \in \{\text{GPi}, \text{STN}\}$, we computed cross-validated fraction of variance explained (FVE) by regressing observed signals $Y^r$ on latent trajectories $Z$:

$$\text{FVE}(Z \to Y^r) = 1 - \frac{\sum \|Y^r - \hat{Y}^r(Z)\|^2}{\sum \|Y^r - \bar{Y}^r\|^2},$$

where $\hat{Y}^r(Z)$ is the ridge-regression prediction and $\bar{Y}^r$ the global mean. We evaluated three cases:

$$\text{FVE}_S^r = \text{FVE}(Z_S^r \to Y^r), \quad \text{FVE}_P^r = \text{FVE}(Z_P^r \to Y^r), \quad \text{FVE}_{SP}^r = \text{FVE}([Z_S^r, Z_P^r] \to Y^r).$$

**Order-free partition.** Using these, we derived an order-invariant variance partition:

$$\text{UniqueShared}^r = \max\big(0, \ \text{FVE}_{SP}^r - \text{FVE}_P^r\big), \qquad \text{UniquePrivate}^r = \max\big(0, \ \text{FVE}_{SP}^r - \text{FVE}_S^r\big),$$

$$\text{Redundant}^r = \max\big(0, \ \text{FVE}_S^r + \text{FVE}_P^r - \text{FVE}_{SP}^r\big).$$

**Model selection rule.** We trained SPIRE across $(d_s, d_p) \in \{3, 4, 5\} \times \{2, 3, 4\}$ per subject–hemisphere. Variants were filtered by: (i) redundancy threshold $\max_r \text{Redundant}^r \leq \tau_{\text{red}} = 0.20$, (ii) collapse guard $\min_r(\text{UniqueShared}^r, \text{UniquePrivate}^r) \geq \tau_{\text{uni}} = 0.01$. Among survivors, the variant with lowest validation reconstruction loss was chosen, preferring smaller $(d_s, d_p)$ in ties.

**Results.** Across subjects with chosen dimensions, GPi latents explained $0.302(\pm0.270)$ unique shared and $0.387(\pm0.215)$ unique private variance; STN latents explained $0.401(\pm0.273)$ and $0.403(\pm0.293)$, respectively. Figure A.3 summarizes the chosen models: each subject–hemisphere shows GPi and STN variance partitioned into UniqueShared, UniquePrivate, and Redundant components. The cohort was heterogeneous: some hemispheres (e.g., S8_R) were private-dominated, while others (e.g., S3) exhibited balanced shared and private contributions. Example UMAP and time-domain traces for both cases are shown in main Fig. 4. Full dimensionality sweeps appear in Fig. A.4.

### A.3.5 TRAINING DETAILS

**Architecture.** Same GRU encoder–decoder backbone as in the synthetic case, but with dropout 0.2. Aligners were initialized as identity mappers with convolutional filters set to impulses.

**Training regime.** Models were trained for up to 200 epochs. For the first 60 epochs, aligners remained in identity mode (no convolutional shifts); convolutional alignment was enabled thereafter. Unlike the synthetic setting, no explicit freeze window was used beyond this identity–to–aligner transition.

**Loss scheduling.** Real-data training followed a four-stage schedule tuned for stability of intracranial recordings. - Epochs $< 60$: warm-up with mapper-only alignment, no cross-reconstruction. - Epochs 60–100: enable convolutional alignment, modest self/cross reconstruction. - Epochs 100–140: increase alignment and cross/self terms. - Epochs $\geq 140$: final tightening of alignment, mapper penalty removed. Loss weights are reported in Table A.3

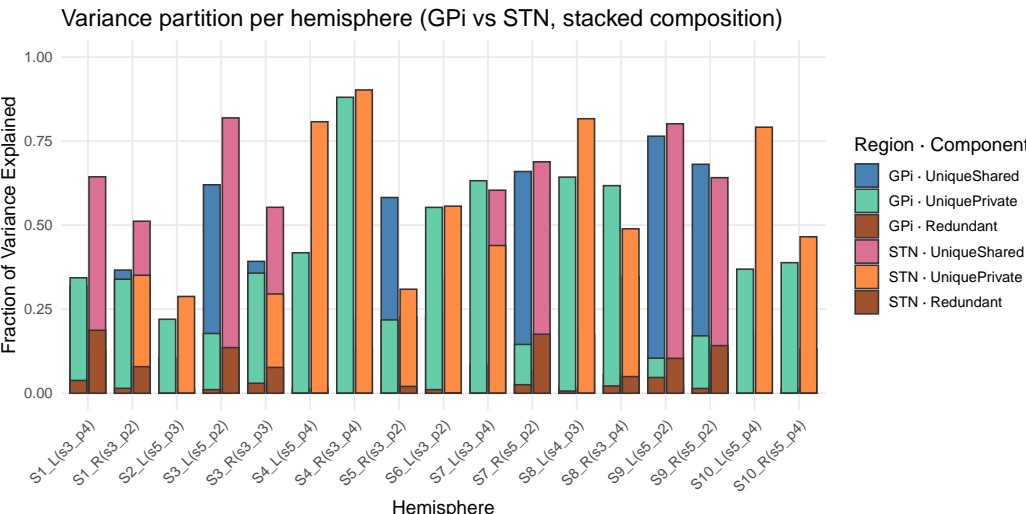

Figure A.3: Subject-specific variance partition of GPi vs. STN. For each subject-hemisphere, two adjacent bars (GPi, STN) show the fraction of variance explained by the *unique shared*, *unique private*, and *redundant* components. Labels include the selected latent sizes $(d_s, d_p)$ per hemisphere chosen via the appendix rule A.3.4). Cohort heterogeneity is evident: several hemispheres (e.g., S8_R) are dominated by private variance (weak GPi–STN shared structure), whereas others (e.g., S3) show a more balanced decomposition consistent with stronger cross-regional coordination. Full dimension sweeps (per hemisphere and region) are in Appendix Fig. A.4.

Table A.3: Loss weight schedule for real DBS training. Values are weight multipliers in the objective.

| Loss component | Epoch ($< 60$) | (60–100) | (100–140) | ($\geq 140$) |
|---|---|---|---|---|
| Reconstruction ($\mathcal{L}_{\text{rec}}$) | 1.00 | 1.00 | 1.00 | 1.00 |
| Alignment ($\mathcal{L}_{\text{align}}$) | 0.30 | 0.30 | 0.38 | 0.45 |
| Orthogonality ($\mathcal{L}_{\text{orth}}$) | 0.012 | 0.012 | 0.015 | 0.015 |
| Self recon. ($\mathcal{L}_{\text{self}}$) | 0.03 | 0.05 | 0.05 | 0.05 |
| Cross recon. ($\mathcal{L}_{\text{cross}}$) | 0.00 | 0.05 | 0.06 | 0.06 |
| Mapper identity ($\mathcal{L}_{\text{mapid}}$) | 0.005 | 0.005 | 0.003 | 0.000 |
| Aligner regularizer ($\mathcal{L}_{\text{align-reg}}$) | 0.000 | $5 \times 10^{-5}$ | $7.5 \times 10^{-5}$ | $1 \times 10^{-4}$ |
| Shared variance guard ($\mathcal{L}_{\text{var}}^{\text{sh}}$) | 0.005 | 0.005 | 0.005 | 0.005 |
| Private variance floor ($\mathcal{L}_{\text{var}}^{\text{pr}}$) | 0.002 | 0.002 | 0.002 | 0.002 |

### A.3.6 BASELINE MODELS

To ensure fair comparison, we implemented and trained two representative baseline models:

**Shared-AE** (Yi et al.) is a recently proposed multi-view autoencoder originally developed for identifying shared subspaces between behavioral and neural activity. Shared-AE uses parallel encoders to extract modality-specific latent representations and combines contrastive alignment losses to encourage shared structure while preserving private variability. We selected Shared-AE as a baseline due to its conceptual similarity to our architecture, particularly in its explicit decomposition of shared and private representations. To adapt it for our intracranial neural recordings from GPi and STN, we leveraged their existing decoder architecture designed for neural data (originally used for the "neural" modality in their paper), applying it to both inputs without any structural modifications. As Shared-AE enforces equal dimensionality for shared and private subspaces, we trained models with total latent dimensions of 6, 8, and 10 (split equally). To evaluate sensitivity to loss scaling, we explored multiple weight combinations for the shared alignment and inverse contrastive losses, including (1.0, 0.05, 0.05) and (1.0, 0.01, 0.01). *Importantly, in this implementation the encoder*

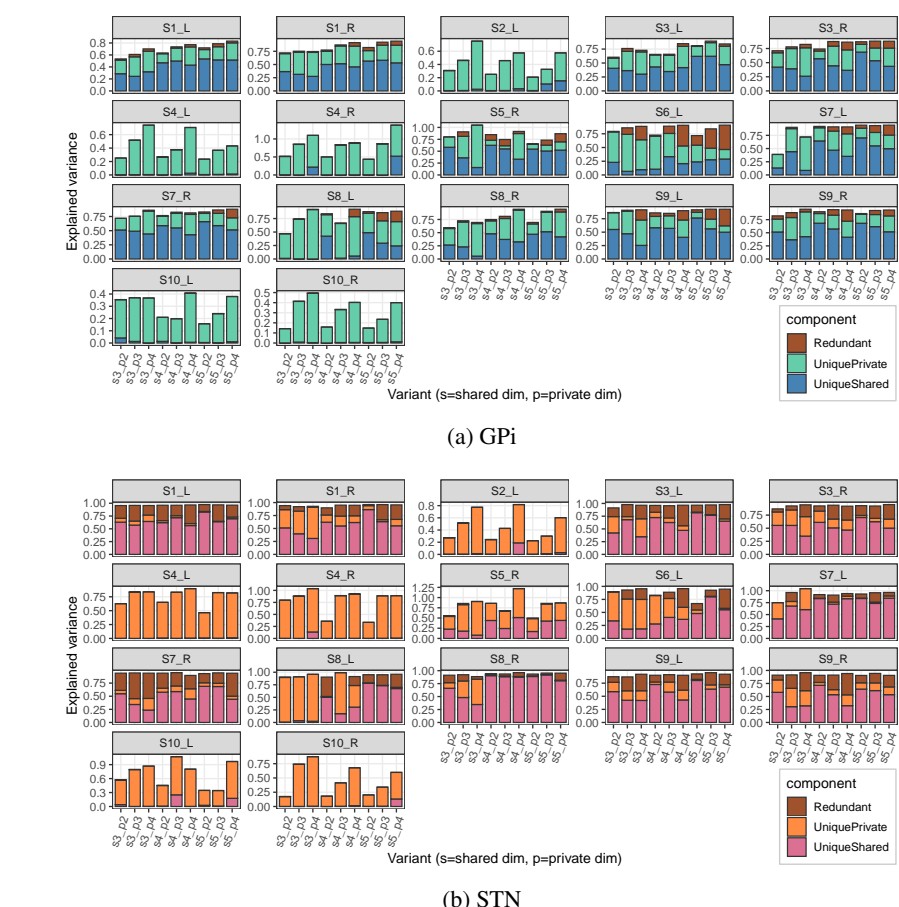

(a) GPi

(b) STN

Figure A.4: Variance partition across latent dimensionalities. For each subject–hemisphere, we trained SPIRE with shared dimension $d_s \in \{3, 4, 5\}$ and private dimension $d_p \in \{2, 3, 4\}$. Each bar shows the fraction of explained variance in the observed signal that is uniquely attributable to shared latents, uniquely attributable to private latents, or redundantly captured by both. (a) GPi recordings. (b) STN recordings. These sweeps were used to select the final $(d_s, d_p)$ configuration per subject–hemisphere, as reported in the main text (Fig. A.3).

*outputs a single embedding vector per window rather than a time-resolved latent sequence. As a result, Shared-AE can be evaluated on reconstruction tasks (including partial reconstructions using shared vs. private vectors), but it cannot be directly applied to analyses that require time-resolved latents, such as CCA similarity against ground-truth trajectories in synthetic data or per-timepoint stimulation classification in human data.*

**Multi-Modal Variational Autoencoder (MMVAE)** (Shi et al., 2019) is a generative framework designed to learn joint latent representations from multiple data modalities (e.g., image–text pairs). It performs modality-invariant inference by averaging per-modality posteriors and optimizing an ELBO objective across all subsets of modalities. We selected MMVAE as a representative baseline for evaluating how well a general-purpose multi-modal generative model can reconstruct brain signals from two interconnected regions (GPi and STN), without relying on explicit disentanglement of shared and private components. To apply MMVAE to multichannel intracranial LFPs, we adapted the encoder and decoder architectures to operate on time-series data by replacing the 2D convolutional blocks with 1D temporal convolutions and transposed convolutions. Each modality-specific VAE encodes multichannel time-series inputs of shape $(C, T)$ into a shared latent space, from which reconstructions are generated. Importantly, we preserved the original MMVAE training paradigm—including the joint inference and reconstruction structure—to ensure a methodologically faithful adaptation to neural signals while enabling a fair comparison against our proposed model that explicitly decomposes shared and private dynamics.

Both baselines were trained on the same 2-region dataset used for SPIRE, using equivalent train/validation splits and comparable model capacities. Training was done using the Adam optimizer with 200 epochs for SharedAE and 100 for MMVAE.

**Fundamental differences.** While both SharedAE and MMVAE provide multi-view latent representations, neither framework is explicitly designed to disentangle shared from private neural dynamics in intracranial recordings. SharedAE aligns subspaces across regions but does not enforce orthogonality or variance partitioning, leading to leakage of region-specific variance into the shared space; moreover, its encoder design precludes extraction of time-resolved latent trajectories. MMVAE assumes mixture-of-experts alignment across modalities, but its architecture lacks explicit mechanisms to separate shared versus private contributions, resulting in latent spaces that cannot support partial reconstructions. In contrast, SPIRE introduces explicit private branches, orthogonality penalties, variance guards, and temporal alignment modules that together enforce a clean separation of cross-regional and region-specific dynamics. These fundamental design choices explain SPIRE's superior performance in human DBS data.

**Attempted DLAG baseline.** We sought to benchmark SPIRE against DLAG (Gokcen et al., 2022), which has been successfully applied to multi-area spiking data. While DLAG worked on our synthetic datasets (Section 4), it proved infeasible on the human intracranial recordings. Specifically:

- Ill-conditioned covariance matrices. After bipolar re-referencing (Appendix A.3.3) and lag augmentation (Appendix A.1.2), the empirical covariance matrices often had deficient rank (e.g., eigenvalues $\approx 0$), causing the Gaussian process kernel inversion in DLAG to fail despite added jitter.

- Model mismatch. DLAG assumes stationary Gaussian processes with smoothly varying latents, whereas intracranial LFPs exhibited strong nonstationarities that violate these assumptions.

- Data dimensionality. Practical channel counts (3–8 bipolar contacts per region) combined with thousands of short windows led to numerical instabilities in the Toeplitz inversion routines central to DLAG. Even under aggressive dimensionality reduction or simplified hyperparameters, the model either diverged or collapsed to degenerate solutions.

We tried multiple remedies—reducing latent dimensionality, adding jitter, trimming channels, and using minimal configurations—but none yielded stable training. We therefore conclude that DLAG, in its current form, is not well suited to our bipolarized intracranial recordings. This limitation highlights the need for alternative frameworks, like SPIRE, that explicitly handle bipolar data, short windows, and nonstationary dynamics.

**Compute comparison**    All experiments were conducted on a local workstation equipped with an NVIDIA RTX A4000 GPU (17.17 GB VRAM), Intel Xeon Silver 4216 CPU (32 logical processors), and 192 GB RAM.

All measurements were made in PyTorch with automatic mixed precision (AMP, fp16) and `torch.inference_mode()`, using batch size $B{=}8$ and inputs with GPi/STN channels $(C_{\text{gpi}}, C_{\text{stn}}){=}(36, 12)$. Each latency is the mean over 50 forward passes after 10 warm-up passes, with device synchronization before/after timing. For MMVAE we use a single-sample ELBO ($K{=}1$) as an inference proxy (one encode–decode pass). SPIRE uses `private_gate=1.0`.

We report two quantities per model (Table A.4): (i) trainable parameters (millions), and (ii) per-sequence inference latency (milliseconds). SPIRE is evaluated by a single call to the model on $(x_{\text{gpi}}, x_{\text{stn}})$. SharedAE is measured as encoders $\rightarrow$ decoder on the same batch. MMVAE latency is measured via $-$ELBO with $K{=}1$ (no gradients).

**Note.** These numbers reflect inference only; training adds backward/optimizer overhead.

**DLAG:** Because DLAG is implemented in MATLAB and relies on an expectation–maximization (EM) procedure rather than a simple forward pass, we measured its compute cost separately. We used synthetic dataset D3 with 8 channels per region and prepared trial-wise sequences as in the DLAG demo code. To obtain an inference-like latency, we timed a single EM iteration (E–M step) with saving disabled using MATLAB's `timeit`, then normalized by the number of sequences. Under these conditions, DLAG required on average 155.85 ms per sequence. We do not report param-

| Model | Params (M) | Seq length $T$ | Inference (ms / seq) |
|---|---|---|---|
| SPIRE (ours) | 0.116 | 247 | 4.45 |
| SharedAE | 23.458 | 244 | 12.22 |
| MMVAE | 0.113 | 244 | 12.74 |

Table A.4: Compute comparison under matched batch size ($B{=}8$), channels $(36, 12)$, and AMP (fp16). Latencies are wall-clock per forward pass (mean over 50 runs; 10 warm-ups). MMVAE uses $K{=}1$ for the ELBO. Note: $T$ differs slightly for SPIRE (247) vs. SharedAE/MMVAE (244) due to preprocessing.

eter counts, since DLAG's MATLAB implementation represents parameters via Gaussian process covariances and delay variables rather than explicit trainable tensors.

### A.3.7 DISTRIBUTIONAL SHIFTS

We measured the Maximum Mean Discrepancy (MMD) between off-stimulation latents and those recorded under each frequency. Unbiased MMD with a Gaussian kernel was computed per latent type, and mixed-effects models with Dunnett-adjusted contrasts tested divergence from baseline. As shown in Figure A.5, GPi stimulation induced significant distributional shifts across all latent types. Both private and shared latents showed increasing divergence with frequency, reaching a plateau at 185–250 Hz. This pattern indicates that GPi stimulation perturbs not only local dynamics but also cross-regional interactions, with the magnitude of disruption scaling with stimulation frequency.

STN stimulation produced a related but more frequency-selective profile. Private STN latents exhibited the largest divergence at 85 Hz, which partially subsided at 185 Hz, suggesting resonance-like tuning of local STN dynamics. In contrast, shared latents continued to diverge significantly at both frequencies, indicating that inter-regional coordination captured by the shared space remains sensitive to increasing frequency. Together, these results show that both shared and private latents are modulated by stimulation, but in complementary ways: private latents reflect site- and frequency-specific perturbations, whereas shared latents encode global, structured signatures that support robust decoding.

Notably, the magnitude of MMD shifts was similar between private and shared latents, yet only shared latents consistently enabled higher classification accuracy (Figure 7). This apparent discrepancy reflects the difference between effect size and separability: private latents may undergo large but variable shifts that reduce discriminability across conditions, while shared latents change in a structured, reliable manner that facilitates accurate decoding. Taken together, the combination of classification and distributional analyses highlights the complementary roles of private and shared latent spaces in capturing stimulation effects.

### A.4 LARGE LANGUAGE MODELS (LLMS) USAGE

We used OpenAI's ChatGPT as a general-purpose assistant for writing/editing, phrasing/organization, and minor LaTeX/code suggestions. The model did not design experiments, analyze data, or generate results; all scientific claims, methods, and code were authored, verified, and validated by the authors. We take full responsibility for the content. The LLM is not an author.

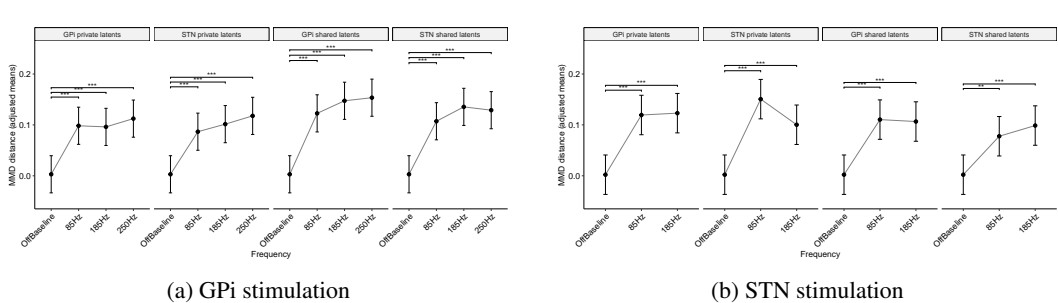

(a) GPi stimulation          (b) STN stimulation

Figure A.5: Distributional shifts induced by stimulation.Maximum Mean Discrepancy (MMD) between off-stimulation latents and those obtained under each frequency, shown as estimated marginal means (EMM $\pm$ 95% CI) from mixed-effects models. (a) GPi stimulation (Off, 85, 185, 250 Hz) induced significant distributional divergence across all latent types, with both private and shared latents showing frequency-dependent increases that plateaued at higher frequencies. (b) STN stimulation (Off, 85, 185 Hz) produced a more selective profile: private STN latents diverged most strongly at 85 Hz and partially subsided at 185 Hz, while shared latents continued to diverge at both frequencies. Stars denote Dunnett-adjusted contrasts vs. OffBaseline (*$p < 0.05$, **$p < 0.01$, ***$p < 0.001$).

