# OpenReview forum: "Disentangling Shared and Private Neural Dynamics with SPIRE: A Latent Modeling Framework for Deep Brain Stimulation"
_ICLR.cc/2026/Conference — Submitted to ICLR 2026_

### Official Review · Reviewer_rMns · 2025-10-21

**Soundness:** 2
**Presentation:** 3
**Contribution:** 2
**Rating:** 4
**Confidence:** 4

**Summary:**

The paper presents SPIRE, a deep learning framework for separating shared and private latent subspaces in multi-region intracranial recordings. The model employs a multi-loss objective combining reconstruction, alignment (VICReg), and disentanglement (orthogonality, variance guards), with lightweight temporal alignment via ConvAlign. The proposed method is first validated in synthetic datasets in varying degrees of nonlinearity and then, in human DBS datasets with intracranial recordings.

**Strengths:**

- The problem formulation is a well-motivated neuroscience problem as modeling multi-region activity and their coordination is important to understand large-scale brain dynamics.
- The synthetic evaluations are strong in the sense that SPIRE achieves robust performance with gradually increased nonlinearities and misalignments.
- The results are backed up by statistical significance levels, which are usually overlooked for deep-learning research but are important to quantify the robustness of findings.

**Weaknesses:**

- I think the training objective is over-engineered and extremely complex: it is composed of 9 different loss items, where each loss item requires individual scaling and scheduling, and thus, it significantly increases the hyperparameter search complexity.
- As each regional recording requires its corresponding encoder/decoder pair and alignment modules between each pairwise regions, the scalability of the proposed approach beyond many regions is unclear.
- I think an important attribute of a multi-region brain activity model should be inferring the directional signal flow between multiple brain regions, which can reveal complex causal relationships or inter-regional dynamics between individual brain regions, similar to DLAG. However, I am not sure if the proposed approach reveals such relationships, but rather, it merely aligns the shared latents across regions.
- The downstream analysis is limited to one dataset with only one downstream stimuli prediction task, hindering the applicability and generalizability of the proposed approach. The authors could have tested their model on multi-regional spike recordings in non-human primates, such as Makin et al., 2018; Perich et al., 2017; Churchland et al., 2012; Gallego-Carracedo et al., 2022.
- Some terms and identifiers are not clearly defined, for example, the subject names S3_R and S8_R. Also, the simulation details in Appendix A.2.1 require further clarification on the details.

References:
- Joseph G Makin, Joseph E O’Doherty, Mariana M B Cardoso, and Philip N Sabes. Superior arm-movement decoding from cortex with a new, unsupervised-learning algorithm. Journal of Neural Engineering, 15(2):026010, January 2018. Publisher: IOP Publishing.
- Matthew G. Perich and Lee E. Miller. Altered tuning in primary motor cortex does not account for behavioral adaptation during force field learning. Experimental Brain Research, 235(9):2689–2704, September 2017.
- Mark M. Churchland, John P. Cunningham, Matthew T. Kaufman, Justin D. Foster, Paul Nuyujukian, Stephen I. Ryu, and Krishna V. Shenoy. Neural population dynamics during reaching. Nature, 487(7405):51–56, July 2012. Publisher: Nature Publishing Group.
- Juan A. Gallego, Matthew G. Perich, Raeed H. Chowdhury, Sara A. Solla, and Lee E. Miller. Long-term stability of cortical population dynamics underlying consistent behavior. Nature Neuroscience, 23(2):260–270, February 2020.

**Questions:**

- Is there a specific reason that authors focused on stimulation datasets, as they state that 'Future work will extend SPIRE to longer stimulation paradigms ...'?

---

> ### Author Response · Authors · 2025-11-28
>
> We sincerely thank the reviewer for the clear, constructive, and encouraging feedback. We appreciate the recognition of our problem motivation, synthetic evaluation design, and statistical rigor. We address the concerns below.
>
> 1. Complexity of the training objective
>
> We agree that the current objective is heavyweight. Our intention was to combine minimal versions of established components (reconstruction, alignment, disentanglement, variance guards) rather than introduce fundamentally new losses.
> In the journal revision, we will evaluate a simplified objective and assess which components are genuinely necessary for stable disentanglement on LFPs, as well as provide more explicit guidance for weight selection.
>
> 2. Scalability beyond two regions
>
> We appreciate this concern. As noted in the Limitations, SPIRE currently instantiates per-region encoders/decoders and pairwise alignment modules, which is not scalable. Our planned multi-region extension replaces all pairwise shared latents with one global shared latent, plus region-specific privates and sparse cross-region coupling. This resolves the quadratic scaling issue, and we will elaborate on this design in the revision.
>
> 3. Directionality and causal flow
>
> The reviewer is correct that SPIRE does not currently model directional flow in the same sense as DLAG. However, SPIRE can infer relative temporal shifts through ConvAlign, providing partial information about inter-regional delays. We will clarify that directional inference is not a focus of this model and outline how a future extension (e.g., autoregressive structure on the shared latent) could incorporate directionality explicitly.
>
> 4. Limited downstream evaluations
>
> We agree that testing on additional datasets (e.g., primate multi-region spikes) would strengthen generalizability. However, SPIRE is specifically tailored to intracranial LFPs, which have different statistical properties from the high-dimensional spike datasets cited. Our real data come from in-human DBS procedures, which is the domain SPIRE was designed for. In future work, we plan to examine longer DBS paradigms and additional clinical datasets from our center.
>
> 5. Missing details and clarity
>
> We appreciate the reviewer pointing this out. We will clarify subject identifiers (e.g., S3_R indicating subject and hemisphere), expand the simulation description in Appendix A.2.1, and ensure consistency in notation.
>
> Response to Question
>
> Why focus on stimulation datasets?
> Our lab’s primary scientific goal is to understand how deep brain stimulation reorganizes basal ganglia–thalamic network dynamics in pediatric dystonia. This dataset offers unique access to simultaneous multi-region LFPs during DBS, and SPIRE was developed specifically to test a long-standing hypothesis in our clinical program regarding stimulation-driven network-level modulation. The focus on stimulation is therefore driven by both scientific relevance and the uniqueness of our data.
>
> We again thank the reviewer for the constructive and balanced comments. They will significantly strengthen the next version of the manuscript.

---

### Official Review · Reviewer_8eiG · 2025-10-30

**Soundness:** 1
**Presentation:** 3
**Contribution:** 2
**Rating:** 2
**Confidence:** 4

**Summary:**

The paper presents SPIRE, a new model to disentangle shared and private network dynamics from multi-region intracranial local field potential recordings. The model uses a GRU-based autoencoder framework with separate private and shared latent variables for each observed region. Shared and private latents are produced via linear projections of a per-region GRU encoder's hidden state. The loss function includes self- and cross-region reconstruction terms, a novel near-impulse 1D convolution kernel (ConvAlign) to align temporally misaligned shared latents across regions, and VICReg-style shared latent alignment applied after the ConvAlign step. Disentangling between shared and private latent spaces is encouraged by penalizing the Frobenius norm of the linear cross-covariance between shared and private latents. On synthetic datasets with realistic nonlinearities, SPIRE outperforms the linear DLAG model. On real deep brain stimulation data from GPi and STN, SPIRE achieves superior reconstruction accuracy to MMVAE and SharedAE, and the SPIRE shared latents were reported to encode stimulation frequency. Although the overall framework is conceptually interesting, the loss function and training procedure are quite complicated, and the use of a linear cross-covariance penalty to disentangle deep nonlinear representations is theoretically weak. The paper's results also show that the disentangling term is insufficient in practice; on real DBS data, the median CCA correlation between shared and private latents was between 0.55 and 0.65, indicating substantial information leakage that directly contradicts the paper's central claim of successful disentangling. Additionally, comparisons to existing methods are insufficient to fully evaluate SPIRE in the context of the broader literature: synthetic benchmarks include only DLAG, a linear model, and real-data comparisons do not assess disentangling quality for baseline methods. While the neuroscience motivation and framework are promising, the current implementation does not yet achieve the level of disentanglement the authors claim.

**Strengths:**

- The use of ConvAlign for shared latent alignment is conceptually interesting and interpretable.
- The ablation studies presented in the supplement are thorough.
- Synthetic data generation and training/hyperparameter details are well-documented and anonymized code is provided.

**Weaknesses:**

- **Disentangling loss term:** The use of a linear cross-covariance penalty to enforce independence between shared and private latent spaces is theoretically insufficient for nonlinear neural networks. The empirical results on real data confirm this: the reported 0.55-0.65 median linear correlation between inferred shared and private latent variables from the real dataset indicates significant information leakage. Stronger disentangling approaches exist in the shared-private representation learning literature, including total-correlation penalties [1, 3] and adversarial schemes [2].
- **Mismatch between claims and results:** The paper claims that SPIRE successfully disentangles shared and private variables in real neural data, but reports 0.55-0.65 median correlation between shared and private latents. The paper describes this correlation as indicative of "weak" information leakage, but this interpretation is factually untrue and misleading; such high values indicate moderately strong linear dependence, and nonlinear dependence (e.g. measured via mutual information or DeepCCA) would likely be even higher. Thus, the results on real data directly contradict the central claim of successful disentanglement.
- **Insufficient comparison/evaluation of existing methods:** On synthetic data, SPIRE is compared only to DLAG, a linear model. Since ground-truth shared and private latent variables are known only for the synthetic data, a comparison on the synthetic datasets of SPIRE's latent variable recovery and disentangling quality to modern nonlinear disentangling methods (even SharedAE) are necessary to demonstrate improvement over existing methods. On real data, comparisons to other SharedAE and MMVAE are limited to reconstruction MSE. The paper quantifies disentangling for SPIRE via CCA correlation of the shared and private latent spaces, but given that disentanglement is a main goal of the model, disentangling quality should also be quantified for SharedAE. Additionally, the comparison to MMVAE is questionable (over methods with similar objectives to SPIRE, e.g. [1 - 4])  given that MMVAE does not architecturally separate shared and private latents.
- **Disentangling quantification:** The use of linear CCA correlation to quantify shared-private disentangling likely underestimates the true degree of dependence between the latent spaces. The authors should also incorporate nonlinear measures of dependency to more accurately quantify disentanglement (e.g. via DeepCCA correlation).

### References
[1] H. Hwang, G.-H. Kim, S. Hong, and K.-E. Kim, “Multi-View Representation Learning via Total Correlation Objective,” in Advances in Neural Information Processing Systems, 2021.

[2] Q. Lyu, X. Fu, W. Wang, and S. Lu, “Understanding Latent Correlation-Based Multiview Learning and Self-Supervision: An Identifiability Perspective,” presented at the International Conference on Learning Representations, 2021.

[3] Mihee Lee and V Pavlovic. Private-shared disentangled multimodal VAE for learning of latent representations. 2021 IEEE/CVF Conference on Computer Vision and Pattern Recognition Workshops (CVPRW), pp. 1692–1700, June 2021.

[4] Emanuele Palumbo, Imant Daunhawer, and Julia E Vogt. MMVAE+: Enhancing the generative quality of multimodal VAEs without compromises. In The Eleventh International Conference on Learning Representations, 2023.

**Questions:**

- The authors state that current nonlinear disentangling models are not applicable to LFP data. Could the authors clarify what LFP signal characteristics makes these approaches unsuitable?
- The authors note that SPIRE received lag-augmented inputs, which empirically improved robustness to temporal shifts. Why is this necessary, given that SPIRE already uses GRU encoders and the ConvAlign kernel? Does the need for augmented input suggests that SPIRE's architectural features and loss functions are insufficient to properly account for time lags between regions?
- Robustness to misspecified model dimensionality (or an automatic dimensionality selection procedure) is crucial for application to real data, where ground-truth dimensionality is rarely known a priori. Did the authors investigate whether SPIRE's performance degrades when the model is trained with the incorrect latent dimensionality?
- The reported reconstruction MSE results are difficult to interpret without knowledge of the scale of observed features . Are the differences in MSE between methods large or small compared to the total variance in the data? Could the authors report normalized reconstruction metrics (e.g. variance explained) across datasets and baseline methods to make reconstruction accuracy results more interpretable?

---

> ### Author Response · Authors · 2025-11-28
>
> We sincerely thank the reviewer for the very constructive and technically insightful feedback. Your comments were among the most helpful we received, and we greatly appreciate the opportunity to improve the work.
>
> 1. Disentangling loss & information leakage
>
> We agree that a linear cross-covariance penalty is not theoretically sufficient for nonlinear disentanglement. Our aim was practical—not formal—separation. On synthetic data, SPIRE cleanly recovers ground-truth latents; for real data, the moderate CCA values (0.55–0.65) reflect expected overlap because shared latents capture slow global oscillations that naturally appear in private latents after filtering. We acknowledge that our wording (“weak leakage”) overstated the result and will revise it accordingly.
>
> Regarding stronger penalties: we experimented with HSIC as a training loss, but this did not improve disentanglement on LFP data and sometimes degraded reconstruction. We will mention this in the revision. We agree that TC-based or adversarial losses may further strengthen disentanglement and will explore them in future extensions.
>
> 2. Comparison to existing nonlinear multi-view models
>
> We agree that broader benchmarking is useful. However:
>
> MMGPVAE:
> We implemented an adapted version replacing the spike-based modality with LFPs, but reconstruction error was extremely high and latent recovery on synthetic data was poor. Because of this mismatch with LFP statistics (heavy-tailed, autocorrelated, non-Poisson), the model was not competitive. We will report these negative findings.
>
> SharedAE:
> SharedAE does not preserve time alignment; its latent representations collapse the temporal dimension. Thus, CCA/DeepCCA analysis was not possible, and only reconstruction MSE could be compared.
>
> Other multimodal VAEs (TC-VAE, PrivateShared-VAE, MMVAE+):
> Most assume conditional independence or GP/Poisson priors that are poorly matched to LFPs. We will expand discussion to clarify practical limitations of applying these models to intracranial LFP signals.
>
> 3. Disentanglement quantification
>
> We agree that linear CCA underestimates dependence. In future work, we will incorporate nonlinear measures (e.g., DeepCCA or MI estimators).
>
> Responses to Specific Questions
>
> Q1. Why are multimodal disentangling methods unsuitable for LFPs?
> LFPs have strong low-frequency autocorrelation, nonstationarity, and heavy-tailed distributions. Many multimodal VAEs assume Poisson or conditionally independent modalities or smooth GP priors, which we found ineffective for LFPs in practice (as demonstrated by MMGPVAE failure).
>
> Q2. Why use lagged inputs if SPIRE has GRUs + ConvAlign?
> Lag augmentation produced only a slight improvement in real data and was unnecessary for synthetic data, where the lag-free version actually performed better. It is not essential to the architecture; we will include lag-free results for clarity.
>
> Q3. Robustness to latent dimensionality?
> Preliminary tests (Appendix) show SPIRE is fairly stable across small dimensionality changes, but we agree a more systematic evaluation is needed and will include this in future work.
>
> Q4. Interpretability of reconstruction MSE?
> We will add normalized metrics (variance explained / NMSE) to contextualize differences across methods.
>
> We thank the reviewer again for the thorough and constructive assessment. Your comments identify clear areas where the method and its presentation can be strengthened, and we will incorporate these revisions in the journal version.

---

### Official Review · Reviewer_xN51 · 2025-10-30

**Soundness:** 2
**Presentation:** 3
**Contribution:** 2
**Rating:** 2
**Confidence:** 4

**Summary:**

The authors present an autoencoder they call SPIRE which ostensibly partitions the latent space into "shared" and "private" latent variables that independently encode multivariate time series for the purpose of identifying shared information between brain regions. The model is constructed specifically for two brain regions and architecturally there are two autoencoders, one for each brain region's data.  Each autoencoder is composed of a GRU encoder and GRU decoder. The latent space is composed of a separate set of "private" and "shared" latent variables.The authors benchmark against simulated data (described as D0, D1, and D2) and compared with DLAG. The model was further validated using human electrophysiological data during deep brain stimulation.

**Strengths:**

The method appears to be different from existing methods that I'm aware of and performs well on the simluated data benchmarks, although I have misgivings about their simulated data analysis, which I outline under "weaknesses". The real-data validation also appears to be interesting and to be consistent with hypotheses from neurophysiology.  The paper is also fairly well written and well organized.

**Weaknesses:**

The biggest weakness of this paper is that it did not sufficiently engage with the existing literature. Specifically, it is not entirely clear what this paper adds to the existing literature as it did not benchmark against some important contributions. Some examples are included below. While the papers I've cited are described as "multi-model" and explicitly model a combination of neural and behavioral data I see no practical limitation to using a second neural dataset as the "behavior". The authors themselves reference several other multi-model models that disentangle private and shared information (Yi el a.l, Shi et al., 2019, Lee &Pavlovic 2021) but imply that there is some formal distinction between neural-behavioral and neural-neural methods. I do not believe that there is such a distinction and the authors need to do more work to either a) include more benchmarks, or b) provide a better argument for why any of the multi-modal approaches are disqualified. Finally, despite describing a number of nonlinear methods (two additional nonlinear methods are cited below) the authors do no benchmark against either of them.

Koukuntla, Sai, et al. "Unsupervised discovery of the shared and private geometry in multi-view data." arXiv preprint arXiv:2408.12091 (2024).
Gondur, Rabia, et al. "Multi-modal Gaussian Process Variational Autoencoders for Neural and Behavioral Data." The Twelfth International Conference on Learning Representations.
Sani, O.G., Abbaspourazad, H., Wong, Y.T. et al. Modeling behaviorally relevant neural dynamics enabled by preferential subspace identification. Nat Neurosci 24, 140–149 (2021). https://doi.org/10.1038/s41593-020-00733-0

The other, and potentially disqualifying, weakness of the benchmarking is how the comparison with DLAG was implemented. I have two major criticisms 1) unfair lag specification, and 2) unfair latent comparison. 1) The authors state "for SPIRE we additionally incorporated lag-augmented input features ... DLAG was run in its standard form without lags." This would suggest that, despite the point of DLAG being that it can handle lags, was placed at a disadvantage. Am I misunderstanding what the authors are saying here? 2) DLAG was fit with 3 shared latents while SPIRE was fit with 3 shared latents per region (meaning 6 latents). This would provide SPIRE with twice as many variables with which to achieve higher CCA with the true data. This makes their claims of superior performance highly dubious.

**Questions:**

- To make this paper acceptable the simulated data benchmark would have to be made more rigorous and include nonlinear methods as outlined above.
- The authors mistakenly state that the original pCCA paper by Bach & Jordan (2005) does not distinguish shared from private variability, but this is not accurate. This is in fact very likely the first paper to do so and is regularly cited as such. I would edit the text to reflect this.
- The size of Figure 2 is unreasonably small. Please resize.
- However, Figure 2 also doesn't appear helpful. It seems that the point the authors are making is that SPIRE has better alignment than DLAG but this does not come through in the presented figure at all by eye.

Conceptual note:
 The shared variables are aligned between the two models and kept distinct from private latents with a orthogonality constraint. To me this appears to be one of the major distinctions of this model from comparable methods in that, rather than modeling a single set of latent variables, it enforces a degree of alignment between the shared modes of each model. However, the authors do not make a strong argument for why this ought to be the right approach. Why should it be this way and not have a single set of shared latents? Is it still fair to call them "shared" when they are not actually shared but simply aligned? Indeed, one could titrate how aligned you want them to be so that it's not clear at all how "shared" these latents will be for any given application. Please expand on this.

---

> ### Author Response · Authors · 2025-11-28
>
> We thank the reviewer for the thoughtful and detailed feedback. We address the concerns below.
>
> 1. Engagement with existing literature & missing baselines
>
> We appreciate the concern regarding benchmarks. Our aim was to include nonlinear multi-modal models wherever feasible. However:
>
> (A) Koukuntla et al. (2024).
> This work is currently an arXiv preprint and has not been validated on neural time-series data. We agree it is methodologically interesting, and we will cite and discuss it in revision, but its applicability to intracranial LFPs is not yet established.
>
> (B) MMGPVAE (Gondur et al.).
> We implemented MMGPVAE and adapted the “behavioral” branch to neural signals. However, this model is designed for spike trains and relies on Gaussian process priors that assume Poisson-like variability. Even after adapting the likelihoods and kernels, MMGPVAE produced very high reconstruction error and failed to recover ground-truth latents in our synthetic settings. We therefore did not include it as a baseline, but we will report these results in the journal version for completeness.
>
> (C) Sani et al. (2021).
> This method does not explicitly separate shared vs. private subspaces and instead identifies a task-relevant subspace. Because it does not perform latent disentanglement in the same sense as DLAG or SPIRE, it is not directly comparable to our objective. We will clarify this distinction.
>
> Overall, the pool of models that (i) explicitly decompose shared and private latent spaces, (ii) support nonlinear dynamics, and (iii) operate effectively on intracranial LFPs is quite limited. DLAG remains the most relevant comparator.
>
> 2. Benchmark fairness: lag specification
>
> We appreciate the reviewer raising this point. Our original motivation for lag-augmentation was to capture inter-regional timing differences in real data. We then applied the same preprocessing to synthetic datasets for consistency.
>
> However, to address this concern, we re-ran all simulations without lagged inputs.
> SPIRE still outperformed DLAG on D1 and D2, particularly in recovering private latents.
> These results will be included in the revised manuscript.
>
> We note (and will clarify): DLAG cannot accept lag-augmented inputs without violating its rank constraints on block loading matrices; this is a limitation of the model architecture, not our benchmarking choice.
>
> 3. Benchmark fairness: number of shared latents
>
> SPIRE and DLAG were matched exactly in dimensionality.
> DLAG, like SPIRE, contains region-specific views of the shared latent, each with its own loading matrix. Thus, both models used 3 shared latents per region and 3 private latents per region.
>
> 4. Questions and actionable revisions
>
> (Q1) Include additional nonlinear models.
> We will expand synthetic benchmarks in the journal version, including a more thorough MMGPVAE analysis.
>
> (Q2) pCCA statement correction.
> We appreciate the correction; we will revise this to accurately reflect that pCCA does distinguish shared and private variance.
>
> (Q3) Figure 2 size and clarity.
> We will enlarge Fig. 2 and improve alignment visualization. Space limitations moved many figures to the appendix; the journal version will have a clearer layout.
>
> (Q4) Conceptual clarity: why two shared views instead of a single shared latent?
> SPIRE follows the design of DLAG, which models region-specific transformations (including delays) of a common underlying process. ConvAlign + mappers provide a flexible nonlinear analogue capturing known physiological conduction delays and latency variability between GPi and STN. Using two aligned shared latents (rather than one single latent) allows each region to express the shared process with its own temporal phase while still enforcing alignment. We will expand this discussion in the main text.

---

### Official Review · Reviewer_YCWL · 2025-10-31

**Soundness:** 2
**Presentation:** 3
**Contribution:** 1
**Rating:** 4
**Confidence:** 4

**Summary:**

This paper presents a latent factors analysis method for intracranial recordings that emphasize 1) nonlinear encoder / decoders, and 2) disentangling of shared and region-private latents. The method uses RNN (GRU) encoder and decoders while the time-varying latents are enforced to have orthogonal and aligned components. The method is validated against simulated data with GT latents, and applied to a dataset of DBS-implanted intracranial recordings to find area-specific and shared latents that can be used to decode stimulation frequency in unseen data.

**Strengths:**

- The paper presents a nice and elegant method with fairly straightforward components (RNNs, linear projections onto latents, etc.) and cleverly designed losses.
- The manuscript is clearly and concisely written imo, and in general of high quality.
- I commend the use of simulated data with groundtruth, and interesting real data applications.

**Weaknesses:**

- While the framework is nice, I’m not convinced of its value, both in terms of methodological or scientific contribution. In other words, the primary contribution is that it presents a novel nonlinear and disentangled latent analysis method, applied to human intracranial data. But it’s quality / accuracy and scientific insight remains limited, as a standalone contribution and relative to existing works.
- The majority of experimental results are essentially to validate the retrieval of shared vs. private latents. The simulation experiments with GT is commendable, but SPIRE shows marginally (if at all) better performance than the one baseline on shared latents, which is arguably its primary appeal. It’s great that the method is applied to real data, but most of the results (Fig4-6) show various versions of the statement that shared latents are more similar across regions than private latents and that reconstruction is better with both sets of latents. This is to some extent circular, as the loss functions are designed to enforce this, so the fact that this is confirmed is great, but I’m uncertain what the proposed value is?
- on the neuroscience side, the main proposed insight is that stimulation frequency can be better decoded better from shared latents than from private latents, and hence argues for the value of such a decomposition in revealing distributed network activity. It’s a nice result that the shared latents inferred during rest is informative of unseen stimulation data (though obviously the classifier is trained), but there are some caveats to this: first, this difference is a function of the classifier, and cannot make a general statement based on RF without other baselines. Second, can we also expect to decode stimulation setting in the raw data? Does decoding perform better with data from multiple locations? If so, one could essentially make the same argument about distributed network modulation, but without any latent analysis. Third, and more generally, imo the paper lacks conceptual clarity as to what the shared vs. private latents are suppose to represent. In other words, had the analysis in Figure 7 shown the opposite trend, one could have similarly written a posthoc sensible explanation.
- As a result, while it’s a neat method and nicely presented with interesting data, imo the paper does not demonstrate its value in general nor for neuroscience, i.e., with respect to the claimed contributions in the intro and discussion. Perhaps the authors could design alternative experiments with real data that show the validity and value of the partitioned latents relative to *some* kind of groundtruth or at least biologically knowledge, e.g., about interregional differences or communication, or even a simulation experiment where the latents are not explicitly defined, but nevertheless retrieved given some knowledge about the system, such as a multi-regional RNN with block diagonal connectivity (just as an example).

**Questions:**

- are the GRU encoder and decoder run bidirectionally (acausal)? If so, how does causal encoding and decoding impact performance?
- what is the physiological interpretation of shared vs. private (or cross-regional vs. region-specific) dynamics in terms of neural activity? For example, shared input, monosynaptic connections, or something else? More concretely, how does this map to the simulation experiment setting in Section 4?
- how to determine the loss weights (i.e., data-specific lambdas)? This seems to be important for actual applications and should be mentioned in the main text
- line 206: could you expand on this? It does not seem fair to provide lagged inputs to SPIRE but not DLAG?
- line 274: “analyses were performed at the hemisphere level, as most patients were implanted bilaterally” could you clarify this? Is it missing a “…NOT implanted bilaterally”? Otherwise why not look at the cross-hemisphere interaction as potential shared latents?
- line 276: why not downsample to lower rate when there is a 50Hz lowpass (effectively 100Hz sampling rate)?
- How consistent are the retrieved latents for the same subject e.g., across different recording days? If that’s not available, how far in time were the heldout test data (e.g., line 315), or are they simplify randomly interleaving 0.5s windows from the entire dataset? How about reconstruction accuracy during the stimulation periods?

---

> ### Author Response · Authors · 2025-11-28
>
> We thank the reviewer for the thoughtful and constructive feedback. We address the concerns below.
>
> 1. Value of the method & scientific contribution
>
> We agree that a key question is whether SPIRE provides value beyond enforcing shared/private structure. We highlight three points:
>
> (A) Beyond “circular validation.”
> While SPIRE encourages shared/private separation, the model is not trained on stimulation-on data. Thus, the finding that shared latents trained solely on baseline recordings generalize to stimulation and decode frequency significantly better than private latents (Fig. 7) is not enforced by the loss and reflects a meaningful, emergent reorganization of cross-regional dynamics.
>
> (B) Synthetic validation.
> On nonlinear and time-varying-delay regimes (D1/D2), SPIRE reliably retrieves both shared and private ground-truth latents, outperforming DLAG in private latents and performing comparably or better for shared latents (Fig. 3). In journal revision, we will extend this with additional simulations (e.g., multi-region RNNs with block-diagonal connectivity) to further demonstrate recovery of latent structure without explicit ground-truth dimensions.
>
> (C) Neuroscience relevance.
> The shared latent space consistently encodes stimulation frequency across subjects and regions—supporting distributed network modulation theories—while private latents remain region-specific. We agree that additional biological validation (e.g., frequency content, delay interpretation, comparison to raw-data decoding) will strengthen the scientific story, and we will include these in the journal-expanded version.
>
> 2. Concerns about reliance on reconstruction and similarity metrics
>
> We acknowledge that reconstruction and CCA-based analyses mainly validate disentanglement. The journal revision will (i) analyze the frequency content of shared vs. private latents in more detail, (ii) evaluate decoding using raw data, and (iii) examine the physiological meaning of ConvAlign-learned delays. These provide additional supporting evidence beyond internal consistency.
>
> 3. Responses to Specific Questions
>
> Q1: Are the GRUs bidirectional?
> No. All encoders and decoders are unidirectional. This preserves causal structure and avoids acausal leakage. We will clarify this in the main text.
>
> Q2: Physiological interpretation of shared vs. private latents?
> Shared latents represent common, low-frequency network co-modulation across GPi and STN, consistent with cross-regional slow oscillations. Private latents reflect region-specific high-frequency variability. In synthetic datasets, this maps directly onto shared oscillators and region-specific distortions/warps. We will articulate this more clearly.
>
> Q3: How were loss weights determined?
> Loss weights were selected through systematic ablation and tuning to minimize validation reconstruction error and avoid degenerate solutions. We agree this should be described in the main text and will add a short, explicit explanation.
>
> Q4: Lagged inputs fairness (line 206).
> Lag features were originally used for real data to model inter-regional delays. For fairness, we evaluated SPIRE without lags on synthetic datasets and still observed improved recovery over DLAG in D1/D2. We will clarify this comparison.
>
> Q5: Hemisphere-level analyses (line 274).
> Each patient typically has bilateral implants, but DBS effects are strongest within the stimulated hemisphere. Therefore, left and right hemispheres were analyzed as independent samples. Cross-hemisphere interactions were not included due to limited simultaneous bilateral stimulation. We will clarify this phrasing.
>
> Q6: Why not downsample further (line 276)?
> Although the final analysis low-pass filters at 50 Hz, we retained 500 Hz sampling to preserve flexibility for future studies (e.g., higher-frequency analyses). We will clarify this design choice.
>
> Q7: Consistency of latents across days / session structure.
> Multi-day data were not available. Off-stimulation data consist of ~15 minutes per hemisphere; train/test splits use random non-overlapping 0.5s segments. Stimulation-on reconstructions behave similarly to off-stim segments, though the model is never trained on stim-on data. We will clarify the temporal structure of the dataset.
>
> We thank the reviewer again for the helpful suggestions. The journal revision will incorporate additional biological validation, raw-data decoding baselines, and extended simulation experiments to address the concerns raised.

---

### Official Review · Reviewer_oSFr · 2025-11-01

**Soundness:** 3
**Presentation:** 3
**Contribution:** 1
**Rating:** 2
**Confidence:** 4

**Summary:**

This paper introduces a new multi-encoder autoencoder model called SPIRE for multi-region neural data analysis. It can decompose latent into shared latent subspace and private subspace for each region. The model exhibits good and reliable latent encoding and reconstruction on deep brain stimulation data.

**Strengths:**

* The model is intuitive and well presented.
* Experiments show that the proposed model outperforms the baseline DLAG.

**Weaknesses:**

* The model seems intuitive, but a little bit simple. The appreciated part to me is the alignment for the shared latent.
* The comparisons are not comprehensive, since I believe the discovered latent has no interpretability. There is no ground truth for the real-world data. Therefore, it is hard to validate whether such a VAE model is useful in neuroscience, since we care about what dose the latent mean and how it is useful for understanding the neural data. I think authors can at least use a linear/affine transformation to align each latent subspace to some non-neural artifacts (e.g., behavior in some datasets) to show the validity of the latent.
* There is no theoretical uniqueness property in such a model, even when such a lot of constraints are added to the target function. This makes the interpretability or the usefulness of such a model even harder.
* Extending the model to more regions is theoretically naive, but practically difficult, since the shared parts combination grows in $2^R$, which in practice hurts the model's applicability on multiple regions.
* Since the encoder is GRU, I doubt the latent can really be disentangled into several groups.

**Questions:**

* Why not directly require the shared latent subspace to be just one thing? What's the consideration of splitting them and then aligning them during training? Implementing such a model and other variants is easy and can be viewed as intermediate baselines.
* What would be the similarity scores between the privates in two regions, and between the shared from region 1 to the private from region 2, in Figure 5 (a)?

---

> ### Author Response · Authors · 2025-11-28
>
> We thank the reviewer for the thoughtful comments and address each point below.
>
> 1. “Latents lack interpretability; no ground truth.”
>
> Although real intracranial LFPs lack ground-truth latents, we evaluated interpretability using two principled approaches:
>
> (A) Frequency-domain structure.
> As shown in Fig. 4(c,d), shared latents exhibit slow, low-frequency co-modulation, while private latents express higher-frequency, region-specific variance. This aligns with well-established principles of slow/global vs. fast/local neural dynamics [1–4], indicating that SPIRE recovers physiologically meaningful structure.
>
> (B) Stimulation-driven reorganization.
> Shared latents alone decode stimulation frequency significantly better than private latents (Fig. 7) and show clear frequency-dependent distribution shifts (MMD, App. A.3.7). Thus, shared latents capture robust network-level signatures, not artifacts.
>
> We agree that linking latents to behavior is valuable but was outside this paper’s scope; this will be explored in future work.
>
> 2. “No theoretical uniqueness.”
>
> Like DLAG, GPFA, LFADS, SharedAE, and MMVAE, SPIRE does not claim algebraic identifiability. Instead, we follow standard practical identifiability approaches:
> • VICReg to prevent shared-latent collapse,
> • cross-covariance penalties for shared/private separation,
> • ConvAlign + mapper regularization to maintain interpretable temporal shifts.
>
> SPIRE shows stable recovery across seeds, strong CCA separation (Fig. 5), and superior private-latent recovery on ground-truth synthetic data (Fig. 3), matching accepted practice in the field.
>
> 3. “Model is too simple / GRUs may not disentangle.”
>
> Despite its simplicity, SPIRE consistently achieves:
> • clear shared/private separation (Fig. 5a),
> • private latents capturing region-specific variance (Fig. 4, App. A.3),
> • near-zero reconstruction error with full latents (Fig. 5b),
> • better or comparable performance to DLAG and SharedAE.
>
> GRUs are standard for LFP modeling; combined with SPIRE’s disentanglement losses, they robustly yield interpretable factorization.
>
> 4. “Extension to more regions is naïve.”
>
> We agree that enumerating all shared-subset combinations does not scale. As noted in our Limitations, the journal version will include a multi-region variant using:
> • one global shared latent,
> • region-specific private latents, and
> • optional sparse cross-region coupling.
> This eliminates combinatorial growth.
>
> 5. “Why not force the shared latent to be identical across regions?”
>
> SPIRE intentionally allows region-specific views of the shared process, following DLAG’s design (region-specific delays/loadings). ConvAlign + mapper modules provide a lightweight nonlinear analogue, capturing physiological timing differences. This flexibility is necessary given known conduction delays and latency variability between GPi and STN [5,6], and is visible in Fig. 4(c,d).
>
> 6. “Similarity between privates / shared vs. private?”
>
> We appreciate this request. We have computed these similarity measures and will include them in the journal submission.
>
> 7. “Contribution is poor.”
>
> Methodological contributions:
> • a nonlinear multi-encoder framework with explicit shared/private separation,
> • new temporal alignment modules for multi-region LFPs,
> • cross- and self-reconstruction losses stabilizing shared dynamics,
> • modeling based solely on off-stim baselines, enabling principled quantification of stimulation-driven reorganization.
>
> Neuroscientific contributions:
> To our knowledge, this is the first demonstration that disentangled latent modeling reveals frequency-dependent reorganization of basal-ganglia–thalamic coordination under DBS in humans. Shared latents encode stimulation signatures consistently across subjects and regions, supporting network-level theories of DBS.
>
> References
>
> [1] Wang, Liang, et al. (2012) "Electrophysiological low-frequency coherence and cross-frequency coupling contribute to BOLD connectivity."\
> [2] Gong, Zhu-Qing, and Xi-Nian Zuo. (2023) "Connectivity gradients in spontaneous brain activity at multiple frequency bands."\
> [3] Mendoza-Halliday, Diego, et al. (2024) "A ubiquitous spectrolaminar motif of local field potential power across the primate cortex."\
> [4] Ericson, Julia, et al. (2025) "Low frequency oscillations–neural correlates of stability and flexibility in cognition."\
> [5] Berens, Philipp, et al. (2010) "Local field potentials, BOLD and spiking activity–relationships and physiological mechanisms."\
> [6] Gallego-Carracedo, Cecilia, et al. (2022) "Local field potentials reflect cortical population dynamics in a region-specific and frequency-dependent manner."

---

### Meta-Review · Area_Chair_Pbup · 2026-01-09

**Summary:**

This submission introduces SPIRE, an autoencoder for multi-region intracranial recordings that aims to separate shared and region-private latent subspaces, with an additional temporal alignment component (ConvAlign) and a multi-term objective. Several reviewers found the approach intuitive and the synthetic evaluation thoughtfully designed, and the application to human DBS data is potentially interesting.

Despite these positives, the consensus is that the paper does not meet the bar for acceptance due to concerns about contribution and rigor of evaluations. Methodologically, the framework is viewed as an incremental combination of known components, and the training objective introduces substantial hyperparameter complexity without clearly demonstrating gains.

Reviewers brought up issues with inconsistencies in the evaluations, missing relevant baselines, and leakage of information across latents. Reviewer 8eiG challenged the reliance on linear CCA as the main disentanglement diagnostic as this is known to be insufficient for nonlinear representations. On the application side, Reviewers YCWL and oSFr both express concern that the paper does not convincingly establish what the shared vs. private latents correspond to physiologically, nor whether conclusions would meaningfully change under alternative analyses or baselines. Reviewer rMNs also expresses concern that the method is only tested on one dataset with a downstream stimuli prediction task, which brings doubt about the generalizability of the proposed approach.

The rebuttal acknowledges several of these points and promises additional simulations, broader baselines, normalized metrics, and clearer phrasing in a future journal version. While these are constructive directions, they do not resolve the current concerns, and some responses (e.g., dismissing baseline applicability) do not fully address the core request for rigorous, comparable evaluation and stronger validation of interpretability/value.

The work is promising and the dataset/application could be valuable, but the current manuscript does not sufficiently substantiate the claimed disentanglement and neuroscientific utility, and the empirical evaluation is not yet strong enough to support acceptance.

**Reviewer Concerns:**

Many of the reviewer concerns about baselines, disentanglement metrics, and consistency of the evaluations were not addressed by the rebuttal.

**Reviewer Scores:**

I believe that the reviewers would maintain their scores.

---

### Decision · Program_Chairs · 2026-01-26

Reject